# Combining Machine Learning Defenses without Conflicts

**Vasisht Duddu**                                     *vasisht.duddu@uwaterloo.ca*
*University of Waterloo*

**Rui Zhang**                                          *zhangrui98@zju.edu.cn*
*Zhejiang University*

**N. Asokan**                                          *asokan@acm.org*
*University of Waterloo*

**Reviewed on OpenReview:** *https://openreview.net/forum?id=C7FgsjfFRC*

## Abstract

Machine learning (ML) models require protection against various risks to security, privacy, and fairness. Real-life ML models need simultaneous protection against multiple risks, necessitating combining multiple defenses *effectively*, without incurring significant drop in the effectiveness of the constituent defenses. We present a systematization of existing work based on *how defenses are combined*, and *how they interact*. We then identify unexplored combinations, and evaluate combination techniques to identify their limitations. Using these insights, we present, DEF\CON, a combination technique which is (a) *accurate* (correctly identifies whether a combination is effective or not), (b) *scalable* (allows combining multiple defenses), (c) *non-invasive* (allows combining existing defenses without modification), and (d) *general* (is applicable to different types of defenses). We show that DEF\CON achieves 90% accuracy on eight combinations from prior work, and 86% in 30 unexplored combinations which we empirically evaluated.

## 1 Introduction

Machine learning (ML) models are susceptible to a wide range of risks to security (Papernot et al., 2018; Tian et al., 2022), privacy (De Cristofaro, 2020; Hu et al., 2022), and fairness (Mehrabi et al., 2021; Pessach & Shmueli, 2022). Several defenses have been proposed to mitigate them. Real-life models require simultaneous protection against multiple risks. But defenses designed to protect against one risk (Li et al., 2023a; Machado et al., 2021; De Cristofaro, 2020; Mehrabi et al., 2021) may impact susceptibility to other unrelated risks (Duddu et al., 2024a). This raises the question of how to combine defenses *effectively*, without incurring a significant drop in the protection provided by each constituent defense. Practitioners need effective *combination techniques*, either by *modifying existing defenses* (invasive), or by identifying whether *existing defenses can be combined without modification* (non-invasive). Prior work is either limited to specific defenses (e.g., Szyller & Asokan (2023); Chen et al. (2023); Fioretto et al. (2022); Noppel & Wressnegger (2024)), or study interactions among defenses with risks (Duddu et al., 2024a; Gittens et al., 2022). No prior work provides a way to quickly determine if *defenses can be combined effectively*.

We first systematically survey existing work on combining defenses based on: (a) *how defenses are combined* (i.e., what combination technique was used), and (b) *how they interact* (i.e., whether they conflict or align). We then identify previously unexplored combinations, and evaluate prior combination techniques to identify their limitations.

Non-invasive combination techniques are easier to deploy as they do not require expert knowledge from practitioners. Therefore, we identify "*mutually exclusive placement*" (Yaghini et al., 2023) as a promising technique. It presumes that two defenses can be effectively combined iff they *operate on different stages in the ML pipeline*: pre-, in-, and post-training. However, it can result in ineffective combinations: (a) later-stage

defenses can conflict with earlier ones (e.g., model or dataset watermarking with adversarial training or differential privacy (Szyller & Asokan, 2023)), and (b) same-stage defenses may still be compatible (see §7).

Based on these insights, we present DEF\CON, a technique which is (i) *accurate* (correctly identifies whether a combination is effective or not), (ii) *scalable* (allows two or more defenses to be combined), (iii) *non-invasive* (does not require changes to the defenses being combined), and (iv) *general* (applicable to different types of defenses). DEF\CON is inspired by mutually exclusive placement (aka naïve technique), but overcomes its limitations by explicitly addressing the reasons that underlie conflicts among defenses: a later-stage defense either (a) mitigates a risk re-purposed as a defense by an early-stage defense, or (b) overrides changes made by an early-stage defense (Szyller & Asokan, 2023). We claim the following contributions:

1. a *systematization* of prior work based on combination techniques, and types of resulting interactions; (§4)
2. *identifying* unexplored combinations, and *evaluating* prior techniques for limitations.; (§5.1 and §5.2)
3. DEF\CON[1], a scalable, non-invasive, and general combination technique (§5.3), which is more accurate than the naïve technique, with a balanced accuracy of
   - 90% (DEF\CON) vs. 40% (naïve) using eight combinations from prior work as ground truth, (§7.2),
   - 86% (DEF\CON) vs. 36% (naïve) via empirical evaluation of 30 unexplored combinations (§7.3 and 7.4).
   DEF\CON constitutes an inexpensive and fast technique for practitioners to determine if a combination of existing defenses without modification can be effective, without using expensive empirical evaluation.

## 2 Background: ML Notations

Consider $X$ as the space of all possible input data records (e.g., images, text prompts) and $Y$ as the space of corresponding outputs (e.g., classification labels for classifiers, predicted next tokens for generative models). An ML model is a function $f^\theta$ which maps $x$ to $y$, i.e., $f^\theta : X \to Y$ where $\theta$ indicates the model's parameters. Hereafter, we denote $f^\theta$ by simply writing $f$. We consider two datasets of the form $(x, y)$, where $x$ is the input data record and $y$ is the output, for training an ML model using training dataset $(\mathcal{D}_{tr})$ and evaluate the model on test dataset $(\mathcal{D}_{te})$. We focus our evaluation (§7) on classifier models, and describe the training and inference for classifiers.

**Training.** We iteratively update $\theta$ using $(x, y) \in \mathcal{D}_{tr}$ over multiple epochs to minimize some objective function $\mathcal{C}$: $\min_\theta l(f(x), y; \theta) + \lambda R(\theta)$ where $l(f(x), y)$ is the prediction error on $x$ for the ground truth $y$. $R(\theta)$ is the regularization function which restricts $\theta$ from taking large values and $\lambda$ is a hyperparameter to control regularization. The parameters are updated as: $\theta := \theta - \alpha \frac{\partial \mathcal{C}}{\partial \theta}$ where $\alpha$ is the learning rate.

**Inference.** We measure the utility of $f$ using its accuracy on $\mathcal{D}_{te}$ computed as

$$\phi_u(f, \mathcal{D}_{\text{test}}) = \frac{1}{|\mathcal{D}_{\text{test}}|} \sum_{(x,y) \in \mathcal{D}_{\text{test}}} \mathbb{I}\left\{ \hat{f}(x) = y \right\}$$

where $\hat{f}(x)$ is the most likely class. If $\phi_u$ is acceptable, $f$ is deployed to provide predictions for input $x$, represented by $f(x)$ for the probability vector across different classes.

## 3 Framework for Systematization

Given a list of defenses whose combinations are explored (§3.1), we present a framework to systematize prior work based on *how the defenses are combined* (§3.2), and *how they interact* (§3.3). We then discuss the completeness of our framework and how to extend it (§3.4).

### 3.1 Defenses being Combined

We describe various defenses proposed to mitigate risks to ML models in the presence of an adversary ($\mathcal{A}dv$). As part of our systematization (§4), we will later enumerate all possible pairwise combination of these defenses.

---

[1]Link to Code: `https://github.com/ssg-research/combining-defenses`.

**Evasion robustness (EvsnRob)** protects against *evasion*, which forces $f$ to misclassify an input $x$ by adding perturbation $\delta_{rob}$ (aka adversarial examples) (Machado et al., 2021; Madry et al., 2018). Here, $\delta_{rob} = argmax_{\delta_{rob}} l(f(x + \delta_{rob}, y)$ and $||\delta_{rob}|| < \epsilon_{rob}$, where $\epsilon_{rob}$ is a perturbation budget.

**Poison robustness (PoisnRob)** protects against *poisoning* which involves training $f$ on *poisons* which are obtained by either tampering existing data records in $\mathcal{D}_{tr}$ or adding manipulated data records to $\mathcal{D}_{tr}$ to degrade $\phi_u$ (Tian et al., 2022). Alternatively, poisoning for backdoors forces $f$ to incorrectly learn a mapping of some pattern in the poisons, to a target class chosen by $\mathcal{A}dv$. During inference, any data record with that pattern is then misclassified to the target class (Li et al., 2022).

**Model Watermarking (MdlWM)** checks for *unauthorized model ownership*, including *model extraction attacks* where $\mathcal{A}dv$ trains a local *surrogate model* to mimic the functionality of $f$ (Orekondy et al., 2019). MDLWM embeds watermarks in $f$ that transfer to the surrogate model during extraction. If a suspect model's watermark accuracy is above some pre-defined threshold, it is identified as a surrogate.

**Fingerprinting (Fngrprnt)** also checks for *unauthorized model ownership* by generating unique identifiers or *fingerprints* (e.g., adversarial examples, embeddings), for $f$. These fingerprints transfer from $f$ to any surrogate model that are derived from it but are distinct from the fingerprints of independently trained models (Cao et al., 2021; Peng et al., 2022; Lukas et al., 2021; Zheng et al., 2022c; Maini et al., 2021).

**Data watermarking (DtWM)** checks for *unauthorized data use* where $f$ is trained on datasets collected without consent (e.g., face images for facial recognition) (Sablayrolles et al., 2020; Huang et al., 2021; Wenger et al., 2023). DTWM either augments $\mathcal{D}_{tr}$ with watermarks (e.g., backdoors) (Tekgul & Asokan, 2022; Sablayrolles et al., 2020), or selects high-influence samples from $\mathcal{D}_{tr}$ as watermarks (Liu et al., 2022a). For verification, we check whether watermarks were in $\mathcal{D}_{tr}$ using statistical tests (Sablayrolles et al., 2020) or membership inference (Liu et al., 2022a). The difference between MDLWM and DTWM is how a model trained from scratch on $\mathcal{D}_{tr}$ is classified: DTWM flags it for unauthorized data use while MDLWM classifies it as independently trained.

**Differential privacy (DiffPriv)** protects against membership inference (whether a data record was in $\mathcal{D}_{tr}$) (Hu et al., 2022) and data reconstruction (reconstructing data records in $\mathcal{D}_{tr}$) (Fredrikson et al., 2015) by hiding whether an individual's data record was used to train $f$ (Abadi et al., 2016). Given two models trained on neighboring datasets differing by one record, DIFFPRIV bounds the privacy loss (distinguishability in predictions between the two models) by $e_{dp}^\epsilon + \delta$. Here, $e_{dp}^\epsilon$ is the privacy budget and $\delta_{dp}$ is probability where the privacy loss is $> e_{dp}^\epsilon$.

**Group fairness (GpFair)** minimizes *discriminatory behavior* to ensure equitable behavior across demographic groups identified by a sensitive attribute in $x$ (e.g., race or sex) (Mehrabi et al., 2021; Pessach & Shmueli, 2022). GPFAIR is measured using various metrics like accuracy parity, demographic parity (Zafar et al., 2019), equalized odds and equality of opportunity (Hardt et al., 2016).

**Explanations (Expl)** give insights into $f$'s *incomprehensible behavior* (Guidotti et al., 2018) which can be used to detect *discriminatory behavior* (Selvaraju et al., 2017; Kim et al., 2018). Explanations $\gamma(x)$ indicate the influence of different input attributes in $x$ on $f(x)$. There are three main categories: *Attribution-based* (Ismail et al., 2021; Smilkov et al., 2017; Sundararajan et al., 2017); *influence-based* (Koh & Liang, 2017); and *recourse-based* (Wachter et al., 2017). We focus on attribution-based explanations which are popular in prior work on combining defenses, and applicable to various domains (e.g., tabular, image). These explanations require training a linear model in a region around a point of interest $x$ (Ismail et al., 2021; Smilkov et al., 2017; Sundararajan et al., 2017). The coefficients of the linear model for an input $x = (x_1, \cdots x_n)$ with $n$ attributes, constitutes $\gamma(x)$.

We summarize the defenses and their impact on $\phi_u$ in Table 8.

### 3.2 Combination Techniques

Based on our survey described later in §4, we identify two combination techniques which either *modify existing defenses*, or identify whether *existing defenses can be combined without modification*. We mark them as **T1** and **T2** respectively, and describe them as follows:

**T1 (Optimization)** includes game-theoretic formalization, regularization, or constrained equation solving. **T1** incorporates defenses into the objective function (e.g., regularization terms) so that the corresponding defense constraints can be satisfied during training for an effective combination (Xin et al., 2023; Hu et al., 2023a; Bu et al., 2022; Wu et al., 2023; Zhang & Bu, 2022; He et al., 2020; Tran et al., 2022; Ali Mousavi et al., 2023; Benz et al., 2021; Ma et al., 2022; Nanda et al., 2021; Xu et al., 2021b; Sun et al., 2022; Li & Liu, 2023; Lee et al., 2024; Wei et al., 2023; P & Abraham, 2021; Liu et al., 2021; Shekhar et al., 2021; Zhang & Davidson, 2021; Tran et al., 2021b; Liu et al., 2022b; Lowy et al., 2023; Jagielski et al., 2019; Tran et al., 2021a; Yaghini et al., 2024; Ding et al., 2020; Xu et al., 2019a; Zhang et al., 2021; Esipova et al., 2023; Xu et al., 2020; Tran et al., 2023; Lakkaraju et al., 2020; Chen et al., 2019; Li et al., 2023b). This also includes using variants of standard model architectures and algorithms, specifically catered for a particular combination to give better trade-offs among the defenses (Ding et al., 2020; Xu et al., 2019a; Yang et al., 2022; Phan et al., 2019; 2020).

**T2 (Mutually Exclusive Placement)** consists of applying defenses at *different* stages of the ML pipeline— i) pre-training (modifies $\mathcal{D}_{tr}$), ii) in-training (modifies training configuration such as objective function), iii) post-training (modifies inputs or outputs of trained $f$ during inference)—to avoid conflicts (Yaghini et al., 2023; Patel et al., 2022). We later refer to **T2** as the *naïve technique* and use it as a baseline to compare with our proposed technique DEF\CON (§5.3 and §7).

### 3.3 Type of Interactions

Consider two defenses $D_1$ and $D_2$ which protect against risks $Rk_1$ and $Rk_2$ respectively, with $D_2$ is applied after $D_1$. The can interact in one of two ways:

- **Alignment.** $D_1$ and $D_2$ are *aligned* if any of these hold: (i) $D_1$ and $D_2$ do not impact $Rk_2$ and $Rk_1$, respectively; (ii) $D_1$ reduces $Rk_2$, increasing $D_2$'s effectiveness; (iii) $D_1$ generalizes $D_2$, so its effectiveness implies that of $D_2$. Alignment leads to an effective defense combination. When one defense implies the other (case (iii)), applying the first may be sufficient since we get the second for free (e.g., attribute privacy and group fairness (Aalmoes et al., 2022)).
- **Conflict.** $D_1$ and $D_2$ *conflict* if any of the following hold: (i) $D_1$ uses risk $Rk$ (protected by $D_2$), making $D_1$ ineffective; (ii) $D_2$ overrides $D_1$'s changes, making $D_1$ ineffective. Conflict leads to an ineffective combination of the defenses. To avoid conflicts, we need accurate combination techniques.

### 3.4 Completeness of Framework

In §3.1, we identify some ML defenses for analysis. We do not claim that this list is complete. For instance, there are other defenses (e.g., *individual fairness* (Dwork et al., 2012), and *interpretability* (Kleinberg & Mullainathan, 2019)), or defenses specific to models other than classifiers (e.g., language and diffusion models), and settings (e.g., federated learning). We can update the framework (§3.1) to add new defenses and enumerate all its combinations with other defenses (as shown later in Table 1). Similarly, in §3.2, we do not claim that the list of combination techniques is complete, but it covers all the techniques seen in our systematization (§4). New combination techniques can be easily added into our framework, and later used for systematization as shown in §4.1.

## 4 Systematizing Interactions among Defenses

We now use our framework to categorize existing literature. We present our methodology (§4.1), and show the systematization of prior work (§4.2).

### 4.1 Methodology

We enumerate all defense combinations from §3.1, in Table 1. Each combination is represented as a cell in Table 1, for which we indicate related work, combination technique used, and the type of resulting interaction.

For each prior work, we identify the following:
**Combination Technique**: We mark the technique to combine defenses as **T1** or **T2**.
**Type of Interaction**: We mark the interactions as a conflict (Ξ), alignment (Ξ), or unexplored (Ξ).

**Justification.** Prior surveys are limited to specific defenses (e.g., Chen et al. (2023); Fioretto et al. (2022); Noppel & Wressnegger (2024)), or do not cover sufficient details to about combination techniques (e.g., Gittens et al. (2022); Ferry et al. (2023)). This makes it challenging to design better combination techniques, and systematically compare with prior works. Our systematization addresses these limitations by (a) covering multiple defenses and their combinations, and (b) explicitly mapping them to the combination techniques and the type of resulting interactions. As shown later in §5, our systematization helps to identify gaps in existing literature (e.g., unexplored combinations, and limitations of prior techniques). Using the insights from our systematization, we can design and evaluate a new combination technique (§5.3 and §7).

**Selecting Papers for Analysis.** We started surveying papers in Google Scholar using keywords (e.g, "combining <defense 1> and <defense 2>"). We selected all papers including those published in top-tier ML and security/privacy venues (e.g., NeurIPS, ICML, ICLR, AAAI, CCS, S&P), related workshop papers, and unpublished papers on ArXiv. We examined their citations and related work to find other papers. Finally, we used papers from related surveys (e.g., Gittens et al. (2022); Chen et al. (2023); Fioretto et al. (2022); Noppel & Wressnegger (2024); Ferry et al. (2023)) to ensure a comprehensive coverage.

### 4.2 Survey of Prior Work

We describe the defense combinations in the order of appearance along the columns in Table 1.

**EvsnRob + PoisnRob.** EvsnRob suppresses adversarial examples (as outliers) while PoisnRob suppresses poisons (as outliers) in $\mathcal{D}_{tr}$. Hence, their objectives are aligned. Viewing PoisnRob as out-of-distribution (OOD) generalization, we can modify adversarial training by incorporating noise from the new domain to improve domain generalization (Xin et al., 2023). This allows the model to learning robust features for OOD generalization, thereby aligning with PoisnRob (Ξ: **T1**). Hu et al. (2023a) defend against both poisons and evasion using a bi-level optimization (Ξ: **T1**).

**EvsnRob + MdlWM.** Adversarial training, as EvsnRob.In, suppresses the influence of backdoors which are used for MdlWM.Pre (Ξ: **T2**) (Szyller & Asokan, 2023). However, generating adversarial-example based watermarks with a higher $\epsilon_{rob}$ than EvsnRob.In, can result in an effective combination (Ξ: **T2**) (Thakkar et al., 2023).

**EvsnRob + DtWM.** Radioactive data (Sablayrolles et al., 2020), as DtWM.Pre, adds backdoors as watermarks to $\mathcal{D}_{tr}$ by perturbing the inputs (similar to adversarial examples). Hence, adversarial training will suppress the influence of watermarks used for DtWM (Ξ: **T2**) (Szyller & Asokan, 2023).

**EvsnRob + Fngrprnt.** Dataset inference (Maini et al., 2021) (as Fngrprnt) is effective with EvsnRob and incurs an acceptable performance drop (Ξ: **T2**) (Szyller & Asokan, 2023). We attribute this to the defenses being applied at different stages (in-training vs. post-training), which reduces conflict between them. On the other hand, a variant of Fngrprnt based on adversarial examples (i.e., "conferrable examples"), are ineffective when EvsnRob is applied for the target or the surrogate model (Ξ: **T2**) (Lukas et al., 2021). We mark them both separately in Table 1.

**EvsnRob + DiffPriv.** Hayes et al. (2022) show that the generalization is worse on combining the objectives of adversarial training (as EvsnRob.In) and DPSGD (as DiffPriv.In), suggesting a conflict (Ξ: **T1**).

Bu et al. (2022) modify the minimax objective function of adversarial training to include DPSGD without violating its guarantees (Ξ: **T1**). Wu et al. (2023) combine randomized smoothing with DPSGD by averaging the gradients of multiple training sample augmentations before clipping to account for the privacy budget of adversarial examples. Both techniques modify the objective function (Ξ: **T1**).

Training $f$ on some public data along with the choice of DiffPriv hyperparameters followed by task specific fine-tuning can result in better trade-off (Ξ: **T1**) (Zhang & Bu, 2022; He et al., 2020). Other works have considered different optimizations: add DiffPriv noise to both input and hidden layers, ensemble adversarial

Table 1: **Overview of Pairwise Combinations among Defenses**: For each combination cell, we citep related work and indicate the "interaction type" (Ξ → alignment, Ξ → conflict, Ξ → unexplored), and "combination technique" used (**T1**-**T2**).

| | EvsnRob | PoisnRob | MdlWM | Fngrprnt | DtWM | DiffPriv | GpFair |
|---|---|---|---|---|---|---|---|
| **PoisnRob** | Ξ→ **T1**: (Xin et al., 2023; Hu et al., 2023a) | | | | | | |
| **MdlWM** | Ξ→ **T2**: (Szyller & Asokan, 2023)

Ξ→**T2**: (Thakkar et al., 2023) | Ξ | | | | | |
| **Fngrprnt** | Ξ→ **T2**: (Szyller & Asokan, 2023)

Ξ→ **T2**: (Lukas et al., 2021) | Ξ | Ξ | | | | |
| **DtWM** | Ξ→ **T2**: (Szyller & Asokan, 2023) | Ξ | Ξ | Ξ | | | |
| **DiffPriv** | Ξ→ **T1**: (Bu et al., 2022; Wu et al., 2023; Phan et al., 2019; 2020; Zhang & Bu, 2022; He et al., 2020) | Ξ→ **T2**: (Xu et al., 2021a; Vos et al., 2023; Ma et al., 2019) | Ξ→ **T2**: (Szyller & Asokan, 2023) | Ξ→ **T2**: (Szyller & Asokan, 2023) | Ξ→ **T2**: (Szyller & Asokan, 2023) | | |
| **GpFair** | Ξ: **T1**: (Tran et al., 2022; Ali Mousavi et al., 2023; Benz et al., 2021; Ma et al., 2022; Nanda et al., 2021; Xu et al., 2021b; Sun et al., 2022; Li & Liu, 2023; Lee et al., 2024; Wei et al., 2023)

Ξ→ **T2**: (Sun et al., 2022) | Ξ→**T1**: (P & Abraham, 2021; Liu et al., 2021; Shekhar et al., 2021; Zhang & Davidson, 2021) | Ξ | Ξ | Ξ | Ξ→ **T1**: (Tran et al., 2021b; Liu et al., 2022b; Lowy et al., 2023; Jagielski et al., 2019; Tran et al., 2021a; Yaghini et al., 2024; Ding et al., 2020; Xu et al., 2019a; Zhang et al., 2021; Esipova et al., 2023; Xu et al., 2020; Tran et al., 2023)

Ξ→ **T2**: (Yaghini et al., 2023) | |
| **Expl** | Ξ→ **T1**: (Lakkaraju et al., 2020; Chen et al., 2019; Li et al., 2023b) | Ξ | Ξ | Ξ | Ξ | Ξ→ **T1**: (Yang et al., 2022); **T2**: (Patel et al., 2022) | Ξ |

training to add adversarial examples to private $\mathcal{D}_{tr}$, or modify objective function for DIFFPRIV guarantees on adversarial examples (Ξ: **T1**) (Phan et al., 2020; 2019).

**EvsnRob + GpFair.** Adversarial training (as EVSNROB.IN) and group fairness (as GPFAIR.IN) have conflicting objectives: EVSNROB.IN pushes the decision boundary away from $\mathcal{D}_{tr}$ while GPFAIR.IN brings it closer (Tran et al., 2022). Also, EVSNROB.IN increases the disparity among demographic subgroups due to class imbalance in $\mathcal{D}_{tr}$ (Hu et al., 2023c) and long-tailed distribution (Lee et al., 2024; Benz et al., 2021; Nanda et al., 2021; Hu et al., 2023c). Several works modify EVSNROB.IN's objective function to ensure equitable model behavior across demographic subgroups by assigning higher weight to minority subgroup (Ξ: **T1**) (Ali Mousavi et al., 2023; Benz et al., 2021; Ma et al., 2022; Nanda et al., 2021; Xu et al., 2021b; Sun et al., 2022; Lee et al., 2024; Li & Liu, 2023). Wei et al. (2023) use different training configurations and assigning different weights to different classes to improve class-wise robustness (Ξ: **T1**).

**EvsnRob + Expl.** Adversarial training (as EvsnRob.In) improves the interpretability of the gradients (Tsipras et al., 2019). This suggesting an alignment with explanations (Chalasani et al., 2020). Also, both defenses can be combined using a minimax objective to construct high fidelity explanations while resisting adversarial examples (Ξ: **T1**) (Lakkaraju et al., 2020; Chen et al., 2019; Li et al., 2023b).

**PoisnRob + DiffPriv.** DiffPriv reduces the influence of outliers thereby improving robustness against poisons as shown in several works (Xu et al., 2021a; Vos et al., 2023; Ma et al., 2019; Jagielski & Oprea, 2021). Hence, DPSGD mitigates poisons and the defenses have aligned objective, resulting in an effective combination (Ξ: **T2**).

**PoisnRob + GpFair.** PoisnRob may overly flag data records from the minority groups as outliers for removal, which increases the bias (Shekhar et al., 2021). This can be corrected by reweighing the scores assigned to outliers to account for sensitive attributes (Ξ: **T1**) (P & Abraham, 2021; Liu et al., 2021). Also, an outlier detector can be trained to minimize the correlation between outlier scores and sensitive attributes (Ξ: **T1**) (Shekhar et al., 2021; Zhang & Davidson, 2021).

**MdlWM + DiffPriv.** DPSGD (as DiffPriv.In) reduces memorization of data records in $\mathcal{D}_{tr}$ and reduces the impact of backdoors for watermarking (MdlWM). Hence, MdlWM conflicts with DiffPriv (Ξ: **T2**) (Szyller & Asokan, 2023).

**DtWM + DiffPriv.** Ideally, DPSGD (as DiffPriv.In) suppresses watermarks (in DtWM), suggesting a conflict. However, empirically, DtWM was effective when combined with DPSGD (Szyller & Asokan, 2023). The adversarial example-based watermarks radioactive watermarking (Sablayrolles et al., 2020), were relatively inliers and not suppressed by DPSGD (Ξ: **T2**) (Szyller & Asokan, 2023).

**Fngrprnt + DiffPriv.** Szyller and Asokan (Szyller & Asokan, 2023) found that DPSGD (as DiffPriv.In) and dataset inference (Fngrprnt) do not conflict, though no reason was provided. We attribute this to defenses being applied in different stages, reducing conflict (Ξ: **T2**).

**DiffPriv + GpFair.** DPSGD (as DiffPriv.In) shows disparate behavior over demographic subgroups (Bagdasaryan et al., 2019). Theoretically, it is impossible to design a high utility binary classifier that is private and fair (Cummings et al., 2019; Agarwal, 2021). Several works modify the objective function by using fairness constraints, regularization, and game theoretic optimization (Ξ: **T1**) (Tran et al., 2021b; Liu et al., 2022b; Lowy et al., 2023; Tran et al., 2021a; Jagielski et al., 2019; Yaghini et al., 2024; Mozannar et al., 2020). Yaghini et al. (2023) combine demographic parity regularization with DPSGD, and estimate fairness on a public dataset to avoid consuming extra privacy budget (Ξ: **T1**). Also, functional mechanism adds Laplace noise to the objective function, along with varied noise levels for different subgroups, which reduces discrimination (Ξ: **T1**) (Ding et al., 2020; Xu et al., 2019a). However, this is limited to the convex objective functions (e.g., logistic regression). Esipova et al. (2023) attribute unfairness in DPSGD to the differences in unclipped and clipped gradient directions. Subsequently, several works have used proposed variable gradient clipping to minimize discriminatory behavior while maintaining utility (Ξ: **T1**) (Xu et al., 2020; Tran et al., 2023; Zhang et al., 2021). Yaghini et al. (2023) use PATE framework to apply fairness constraints and DiffPriv noise in the aggregated votes from the teacher's ensemble. Both fairness and privacy are applied in pre-training (Ξ: **T2**).

**DiffPriv + Expl.** The objectives of these defenses are inherently conflicting: DiffPriv hides information to minimize leakage while Expl releases additional information to improve comprehensibility (Banisar, 2011). Yang et al. (2022) train an autoencoder with DiffPriv (functional mechanism). This autoencoder is used to generate data records and compute counterfactuals that satisfy DiffPriv via the post-processing property (Ξ: **T1**). Patel et al. (2022) propose an adaptive DPSGD algorithm to generate high-quality explanations without consuming $\epsilon_{dp}$, by reusing past explanations for similar data records (Ξ: **T2**).

## 5 Insights from Systematization

We identify unexplored combinations (§5.1), requirements for an ideal technique, limitations of prior techniques (§5.2), and design a new technique (§5.3).

### 5.1 Unexplored Combinations

We identify 14 unexplored combinations ($\Xi$ in Table 1): (i) PoisnRob with {MdlWM, DtWM, Fngrprnt, Expl}; (ii) MdlWM with {DtWM, Fngrprnt, GpFair, Expl}; (iii) DtWM with {Fngrprnt,GpFair,Expl}; (iv) Fngrprnt with {GpFair, Expl}; (v) GpFair with Expl.

> **Takeaway**: Unexplored combinations reveal research gaps and opportunities for effective technique design.

We revisit these combinations in our evaluation (§6 and §7).

### 5.2 Evaluating Combination Techniques

Our systematization reveals that some techniques may lead to ineffective combinations. We outline requirements for an ideal technique, and identify limitations in prior work.

**Requirements.** A combination technique should allow practitioner to quickly determine whether a combination can be effective combined. Empirical evaluation to determine the effectiveness of a combination, while definitive, can be expensive, especially when multiple defenses are involved. An ideal combination technique should be: **R1 (Accurate)** correctly identifies whether a combination is effective or not; **R2 (Scalable)** allows two or more to be combined simultaneously; **R3 (Non-invasive)** does not require modifying defenses, easing adoption and removing the need for expert knowledge; **R4 (General)** applicable to various defenses.

> **Takeaway:** Combination techniques, including newly proposed ones, should be evaluated for **R1**-**R4**.

**Limitations of Prior Techniques.** We summarize the limitations of existing techniques (**T1**-**T2**) as per the requirements (**R1**-**R4**) in Table 2. We use ○ for requirement not satisfied, ◑ for partially satisfied, and ● for fully satisfied.

**T1 (Optimization)** where modifying the objective function for training, followed by hyperparameter tuning, can result in an effective combination. However, this often results in a trade-off between effectiveness of constituent defenses and model utility. Hence, we mark **T1** as partially accurate (**R1** → ◑). This trade-off also explains why prior works have struggled to scale beyond two defenses (**R2** →

Table 2: Requirements satisfied by various techniques: ○ → Not satisfied; ◑ → Partially satisfied; ● → fully satisfied.

| Technique | R1 (Accurate) | R2 (Scalable) | R3 (Non-Invasive) | R4 (General) |
|:---:|:---:|:---:|:---:|:---:|
| T1 | ◑ | ○ | ○ | ○ |
| T2 | ◑ | ● | ● | ● |

○). Furthermore, some defenses are not applicable for **T1** (e.g., Expl and Fngrprnt), and require modifications or non-standard variants (**R3** → ○). Finally, these optimizations are tailored to specific defenses being combined, and do not apply to other defenses. They are also specific to some models (e.g., logistic regression with DiffPriv), and do not translate to other models (e.g., neural networks). Hence, **T1** has limited applicability (**R4** → ○).

**T2 (Mutually Exclusive Placement)** can apply up to one defense in each of the three stages, thus, making it scalable (**R2** → ●). Defenses do not need any modification (**R3** → ●), and the combination technique is applicable to all types of defenses (**R4** → ●). However, the combinations may not be effective: (i) a defense in a later stage of the pipeline can conflict with earlier ones (false negatives) (Szyller & Asokan, 2023), and (ii) it rules out defenses in the same stage that do not conflict (false positives see §7). Hence, this may incorrectly identify effective combinations (partially accurate **R1** → ◑).

> **Takeaway.** *Neither technique satisfies all the requirements.* **T2** is promising as it satisfies **R2**, **R3**, and **R4**, but not **R1** (incurs false positives and false negatives).

### 5.3 Def\Con: Design

From our systematization, we identify that **T2** overlooks underlying causes for conflicts, leading to false positives/negatives. *Can we address the limitations of **T2** and improve **R1**?* Recall from §3 that conflicts arise when (i) an early-stage defense uses a risk which is mitigated by a later-stage defense, or (ii) changes

by an early-stage defense is overridden by a later-stage defense. We conjecture that by accounting for these underlying causes, we can meet **R1**. We present DEF\CON, a principled technique to identify effective defense combinations, by accounting for the reasons underlying conflicts

**Methodology to Derive Def\Con.** We start with the naïve technique and modify it to include the underlying causes for conflicts among defenses. We iterate over the design of DEF\CON using prior work from §4, and evaluate the final design on unexplored combinations (see §7).

**Def\Con Description.** We describe DEF\CON using the example of combining two defenses, $D_1$ and $D_2$ which protect against $Rk_1$ and $Rk_2$ respectively, and later discuss how to extend to more than two defenses. Following prior work (Duddu et al., 2024a), we refer to unintended interactions between a defense and a risk if the defense either increases or decreases the susceptibility to an unrelated risk (e.g., $D_1$ and $Rk_2$).

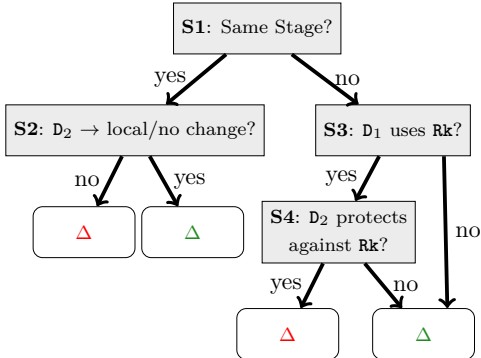

Figure 1: Flowchart depicting various steps in DEF\CON to identify conflict between $D_1$ and $D_2$ ($D_2$ is applied after $D_1$).

We start by identifying variants of each of the defenses across pre-, in-, and post-training stages (see Table 8). We compare each variant of $D_1$ with that of $D_2$, and use $\triangle$ for alignment, and $\triangle$ for conflict. Assuming $D_1$ is applied first and then $D_2$, we follow the steps below *in sequence*:

**S1** Are $D_1$ and $D_2$ applied in the same stage?
• If yes, go to **S2** • Else, go to **S3**

**S2** The type of changes made by the defenses determines whether there is a conflict. We classify the changes as *global*, *local*, and *none*. *Global changes* modify $f$ (e.g., training with a regularization term, pruning) or transform all records in $\mathcal{D}_{tr}$ (e.g., synthetic data generation for DP or fairness during pre-training). *Local changes* affect specific data records (e.g., adding watermarks in pre-training or modifying certain predictions in post-training). FNGRPRNT.POST and EXPL.POST make *no changes* to $f$ and $\mathcal{D}_{tr}$.

• If $D_1$ makes global/local/no changes while $D_2$ makes local/no changes, we mark this as $\triangle$.
  **Rationale:** *Changes by $D_1$ will not interfere with local/no changes by $D_2$, as $D_1$ is applied first. Hence, no conflict.*

• If $D_1$ makes global/local/no changes while $D_2$ makes global changes, mark as $\triangle$.
  **Rationale:** *Global changes by $D_2$ will override changes by $D_1$, thereby reducing its effectiveness. This is called catastrophic forgetting when the defenses are applied sequentially during training (Kemker et al., 2018; Szyller & Asokan, 2023). This is a conflict.*

**S3** $D_1$ and $D_2$ are in different stages. Does $D_1$ use a risk $Rk$ as part of the defense (e.g., watermarking uses backdoors)?

• If yes, go to **S4**.

• If no, mark as $\triangle$.
  **Rationale:** *If $D_1$ does not use $Rk$, the susceptibility to $Rk$ will not be impacted after applying $D_2$. Hence, $D_1$ and $D_2$ are unlikely to interfere with each other.*

**S4** Does $D_2$ protect against $Rk$ either explicitly or via unintended interaction?

• If yes, mark as $\triangle$.
  **Rationale:** *Since $D_1$ uses $Rk$ (either explicitly or via unintended interaction), $D_2$ will reduce susceptibility to $Rk$ making $D_1$ less effective. Hence, there is a conflict.*

• If no, mark as $\triangle$.
  **Rationale:** *$D_1$ and $D_2$ are unlikely to interfere with each other. Hence, there is no conflict.*

We summarize the steps in DEF\CON in Figure 1. DEF\CON evaluates combination effectiveness based solely on the effectiveness of the constituent defenses, without considering the model utility. We revisit model utility in §8. We also present a formal analysis of DEF\CON covering consistency, soundness, and completeness in Appendix B.

**Note on DiffPriv.** Combining DIFFPRIV with other defenses does not consume additional privacy budget. Any modification to $\mathcal{D}_{tr}$ (e.g., adding watermarks) is done within the privacy boundary, and does not consume privacy budget. Any defense applied after DIFFPRIV is "free" (DIFFPRIV's post-processing property).

**Extending Beyond Two Defenses (Multi-way Combinations).** To extend DEF\CON beyond two defenses, our algorithm decomposes the multi-way combination into pairwise comparisons, invoking the original DEF\CON flowchart (Figure 1) to check for conflicts. Determining conflict for multi-way combinations from pairwise combinations is principled and not limiting. We have to consider two possible cases when decomposing multi-way combinations into pairwise combinations:

- **Case 1: Defenses in different stages.** When we combine three defenses in different stages, checking conflicts among all pairwise combinations is fine since the order in which we apply defenses is fixed. Any conflict detected among any pair, will be marked as an overall conflict (marked as $\triangle$).
- **Case 2: More than one defense in one stage.** We have to check all possible permutations of the defenses in a given stage, and determine whether there is a conflict. If there is no conflict, we combine with other stage defenses, and check for the conflicts.

We now present our algorithm to evaluate conflict for multi-way combinations, and assume that we have a set of defenses—partitioned into three ordered stages. We iterate through each stage in order, and skip stages with no defenses:

**M1 If stage has single defense:** For the stage $s$, we consider the defense as $\mathtt{D}_s$ and check its compatibility with defenses in other stages. $\mathtt{D}_s$ is $\mathtt{D}_1$, $\mathtt{D}_2$ or $\mathtt{D}_3$ depending on $s$.
**M2 If stage has multiple defenses:** For each stage with multiple defenses:
  **M2.1 Permutation Checking:** Consider all permutations of the defenses. For each permutation, check every consecutive pair of defenses using DEF\CON flowchart to detect conflicts for two defenses.
  **M2.2 Pruning and Selection**: If a pair in a permutation causes a conflict, discard that permutation and prune others containing that conflicting pair (e.g., for defenses: A, B, and C, if "AB" conflict, then no need to check for "ABC" or "CAB"). If a conflict-free permutation is found, treat the entire sequence as a single composite defense, $\mathtt{D}_s$ (e.g., let us say "CAB" does not conflict).
**M3 Sequential Composition**: Treat the resulting composite defense as an atomic unit (e.g., $\mathtt{D}_1$ = "CAB") and check for conflicts with the next stage's resolved defenses (e.g., $\mathtt{D}_2$ = "DE"), using DEF\CON flowchart between $\mathtt{D}_1$ and $\mathtt{D}_2$. In other words, we check whether any of the constituent defenses in $\mathtt{D}_1$ use a risk which is mitigated by constituent defenses in $\mathtt{D}_2$ (**S3** in Figure 1).
**M4 Termination:** If any invocation of DEF\CON for pairwise check indicates a conflict, terminate and report as a conflict (marked as $\triangle$). Else, indicate as alignment (marked as $\triangle$).

We present a formal analysis of the algorithm to extend DEF\CON for more than two defenses in Appendix C. Having discussed the DEF\CON's design, we comprehensively evaluate DEF\CON across **R1**-**R4**.

# 6 Experimental Setup

## 6.1 Datasets and Models

We use two image datasets: `FMNIST` and `UTKFACE`. `FMNIST` consists of 28x28 grayscale images of ten clothing types, with 60,000 training and 10,000 testing images. We classify these using a two layer CNN with 16 and 32 filters, ReLU activation, and a fully connected layer for ten-class classification. `UTKFACE` includes 48x48 RGB images, classifying individuals as young (under 30), with 11,852 training and 10,667 testing images. It also includes the sex of the individuals as a sensitive attribute. We use a VGG16 model with a fully connected layer for binary classification.

We choose `FMNIST` since all of the defenses we consider for evaluation (§6.3) have used it for evaluation. We selected `UTKFACE` because many defenses effective for `FMNIST` are likely to be applicable to it as well, given that both are image datasets. Also, `UTKFACE` includes sensitive attributes, making it suitable for GPFAIR.

## 6.2 Revisiting Defenses

Since the naïve technique and DEF\CON apply defenses at different ML pipeline stages, we revisit and categorize defenses in §3.1 by the stage they are applied in. For each defense from §3.1, we specify the variants in pre-training ("<defense>.**Pre**"), in-training ("<defense>.**In**"), and post-training ("<defense>.**Post**"). For additional context, we indicate the impact of applying a defense on $\phi_u$ compared to a "no defense" baseline, where "$\vee$" is a decrease, "$\sim$" is no effect, and "$\wedge$" is an increase.

### Evasion robustness (EvsnRob)

- **EvsnRob.Pre (Data Augmentation)** where adding transformations of training data records improves robustness (Yun et al., 2019; Zhang et al., 2018b; DeVries & Taylor, 2017; Rebuffi et al., 2021) (but see §6.3). This improves $\phi_u$ ($\wedge$) by acting as regularization (Yun et al., 2019; Zhang et al., 2018b; DeVries & Taylor, 2017; Rebuffi et al., 2021).
- **EvsnRob.In (Adversarial Training)** modifies the objective function to minimize the maximum loss from adversarial examples (Madry et al., 2018; Zhang et al., 2019): $L_{advtr} = \min_\theta \frac{1}{|\mathcal{D}_{tr}|} \sum_{x,y \in \mathcal{D}_{tr}} \max_{\|\delta\| \le \epsilon_{rob}} \ell(f(x+\delta), y)$. Alternatively, randomized smoothing modifies the training and inference to obtain certified robustness of $f$ (Cohen et al., 2019; Lecuyer et al., 2019). These defenses decrease $\phi_u$ ($\vee$) (Zhang et al., 2019; Tsipras et al., 2019).
- **EvsnRob.Post (Input Processing)** removes adversarial perturbations before passing them to $f$ (e.g., using generative models (Nie et al., 2022; Song et al., 2018) or input encoding (Buckman et al., 2018; Guo et al., 2018; Das et al., 2017)) or checks for adversarial examples using statistical tests (Grosse et al., 2017). Defenses which modify input images using generative models decrease $\phi_u$ ($\vee$) (Nie et al., 2022; Song et al., 2018; Guo et al., 2018; Das et al., 2017). If the input transformation is small, the decrease in $\phi_u$ is negligible ($\sim$) (Buckman et al., 2018; Grosse et al., 2017).

### Poison robustness (PoisnRob)

- **PoisnRob.Pre (Data Sanitization)** includes detecting and removing outliers in $\mathcal{D}_{tr}$ (e.g., using Shapley values (Jia et al., 2021b; 2019; Doan et al., 2020) or anomaly detection (Cretu et al., 2008; Paudice et al., 2018; Tran et al., 2018; Barreno et al., 2010; Chen et al., 2018)), followed by retraining. As the outliers are memorized and contribute to $\phi_u$, their removal degrades $\phi_u$ ($\vee$) (Jia et al., 2021b; 2019).
- **PoisnRob.In (Fine-tuning)** updates $f$ to minimize outlier influence. This includes distillation to reduce the influence of poisons (Li et al., 2017) or fine-tuning on a poison-free dataset (Diakonikolas et al., 2019; Zhu et al., 2023; Xu et al., 2019b; Liu & Guo, 2020; Patrini et al., 2017). These do not impact $\phi_u$ ($\sim$).
- **PoisnRob.Post (Pruning)** reduces the effectiveness of backdoors by removing some model parameters based on the observation that poisoned and clean samples have different activations (Liu et al., 2018; Wu & Wang, 2021; Zheng et al., 2022b;a; Li et al., 2023c). This degrades $\phi_u$ ($\vee$).

### Model Watermarking (MdlWM)

- **MdlWM.Pre (Backdoors)** uses backdoor watermarks in $\mathcal{D}_{tr}$ (Adi et al., 2018; Zhang et al., 2018c; Jia et al., 2021a; Uchida et al., 2017). These are designed to maintain $\phi_u$ ($\sim$).
- **MdlWM.In (Optimization)** updates the original objective function to include watermark behavior (Bansal et al., 2022; Bagdasaryan & Shmatikov, 2021). For instance, certified watermarking adds Gaussian noise to watermarks (added to $\mathcal{D}_{tr}$) to get certification on watermark accuracy (Bansal et al., 2022). Also, backdoors can be introduced through regularization, which can be repurposed for watermarking (Bagdasaryan & Shmatikov, 2021). This degrades $\phi_u$ ($\vee$).
- **MdlWM.Post (API)** modifies predictions to embed watermarks (Szyller et al., 2021) which are used by $\mathcal{A}dv$ to train the surrogate model. These are designed to maintain $\phi_u$ ($\sim$).

**Fingerprinting (Fngrprnt).** All fingerprints are post-training schemes (denoted as **Fngrprnt.Post**). No retraining or modification of $f$ is required and hence, FNGRPRNT has no effect on $\phi_u$ ($\sim$).

**Data watermarking (DtWM).** All the current schemes are during pre-training (**DtWM.Pre**), and are designed to maintain $\phi_u$ ($\sim$). The difference between MDLWM and DTWM is how a model trained from scratch on $\mathcal{D}_{tr}$ is classified: DTWM flags it for unauthorized data use while MDLWM classifies it as independently trained.

**Differential privacy (DiffPriv)**

- **DiffPriv.Pre (Private Data)** from generative models with DiffPriv constraints, that can be used for downstream tasks instead of $\mathcal{D}_{tr}$ (Hu et al., 2023b; Xie et al., 2018; Torkzadehmahani et al., 2019; Zheng & Li, 2023). This decreases $\phi_u$ ($\vee$).
- **DiffPriv.In (DPSGD)** trains $f$ by adding carefully computed noise to the gradients to minimize the influence of individual data records on $f$ (Abadi et al., 2016). Private aggregation of teacher's ensembles (PATE) (Papernot et al., 2017) is another framework for DP where multiple teacher models are trained on disjoint private datasets, while a student model is trained on a public dataset with labels annotated via noisy voting from the teacher models. These defenses decrease $\phi_u$ ($\vee$) (Jayaraman & Evans, 2019).
- **DiffPriv.Post (Output Perturbation)** includes adding calibrated noise to the output of empirical risk minimization objective (Chaudhuri et al., 2011). This decreases $\phi_u$ ($\vee$). The theoretical guarantees are poorer than other DP defenses and DiffPriv.Post requires the objective function to be convex. We omit this since it does not cover neural networks.

**Group fairness (GpFair)**

- **GpFair.Pre (Fair Data)** modifies $\mathcal{D}_{tr}$ to reduce bias in the downstream model (Kamiran & Calders, 2011; Calmon et al., 2017; Zemel et al., 2013; Feldman et al., 2015). This degrades $\phi_u$ ($\vee$).
- **GpFair.In (Regularization)** penalizes violating fairness constraints (Agarwal et al., 2018; 2019; Celis et al., 2019; Kamishima et al., 2012). This degrades $\phi_u$ ($\vee$) (Zhang et al., 2018a; Louppe et al., 2017; Pinzón et al., 2023).
- **GpFair.Post (Calibration)** adjusts the threshold over the predictions to ensure that the prediction probabilities accurately reflect the true likelihood across each demographic group (Pleiss et al., 2017; Hardt et al., 2016; Kamiran et al., 2012; Geyik et al., 2019; Salvador et al., 2022; Kull et al., 2017; Hebert-Johnson et al., 2018) This degrades $\phi_u$ ($\vee$) (Pleiss et al., 2017).

**Explanations (Expl).** Expl are post-training defenses (**Expl.Post**) which does not require retraining, and hence does not degrade $\phi_u$ ($\sim$).

We summarize the defenses in Appendix A: Table 8.

## 6.3 Choosing Defenses for Evaluation

To select defenses for our evaluation, we began with those in §6.2 (summarized in Appendix A: Table 8). We remove defenses which are not robust: EvsnRob.Post (Input Processing) and PoisnRob.Pre (Data Sanitization) (Kang et al., 2024; Koh et al., 2022). We then evaluate the remaining defenses and exclude those which were ineffective on our datasets: EvsnRob.Pre (Data Augmentation) (Yun et al., 2019; Zhang et al., 2018b; DeVries & Taylor, 2017), DiffPriv.Pre (Private Data) (Zheng & Li, 2023), GpFair.Pre (Fair Data) (Zemel et al., 2013), and GpFair.Post (Calibration) (Pleiss et al., 2017). DiffPriv.Pre, GpFair.Pre, and GpFair.Post, were designed for tabular datasets but ineffective on our image datasets. We speculate about these defenses in §8.

We are left with eleven defenses: (i) EvsnRob.In (adversarial training), (ii) PoisnRob.In (fine-tuning), (iii) PoisnRob.Post (model pruning), (iv) MdlWM.Pre (backdoor watermarks), (v) MdlWM.In (watermarks via objective function), (vi) MdlWM.Post (API-based watermarks), (vii) DtWM.Pre (backdoor watermarks), (viii) Fngrprnt.Post (dataset inference), (ix) DiffPriv.In (DPSGD), (x) GpFair.In (regularization), (xi) Expl.Post (attribution). We get 55 pairwise combinations from them but we remove combinations among defenses with the same objective: three combinations among watermarking (MdlWM.Pre, MdlWM.In, MdlWM.Post), three for Fngrprnt.Post with MdlWM.Pre, MdlWM.In, MdlWM.Post, and one for PoisnRob.In and PoisnRob.Post. This leaves us with 48 combinations.

## 6.4 Metrics and Implementations

We describe the metrics for evaluating the effectiveness of each defense, and the implementations taken from publicly available code from prior work. We measure $\phi_u$ on $\mathcal{D}_{te}$ for all defenses. Our implementations for

defenses are based on the state-of-the-art (FNGRPRNT, POISNROB), standard libraries (DIFFPRIV, GPFAIR, EXPL), or seminal work (EVSNROB, DTWM, MDLWM). We use the standard hyperparameters which are either from the literature or the library documentation, such that the resulting individual defenses are effective (Table 3). When combining defenses, we use the same hyperparameters, but revisit hyperparameter tuning for defenses in combination (see §7.4). We report the mean and standard deviation across five runs.

**Evasion Robustness (EvsnRob.In).** We use the accuracy on $\mathcal{D}_{rob}$ which is obtained by replacing data records in $\mathcal{D}_{te}$ with the adversarial variants:

$$\phi_{\text{robacc}}(f_{\text{EvsnRob}}, \mathcal{D}_{\text{rob}}) = \frac{1}{|\mathcal{D}_{\text{rob}}|} \sum_{(x,y) \in \mathcal{D}_{\text{rob}}} \mathbb{I}\left\{\hat{f}_{\text{EvsnRob}}(x) = y\right\}$$

Ideally, $\phi_{\text{robacc}}^{\text{EvsnRob}}$ should be close to $\phi_{\text{u}}$. We use the original implementation of TRADES (Zhang et al., 2019) and an implementation of AutoAttack (Croce & Hein, 2020) in SecML library (Pintor et al., 2022b) from Pintor et al. (2022a). As we evaluate effectiveness of EVSNROB.IN using attacks, poorly optimized attacks can falsely suggest defense effectiveness (Carlini et al., 2019; Carlini & Wagner, 2017; Tramer et al., 2020). We individually optimize these attacks for evaluation with defenses and their combinations, following Pintor et al. (2022a) to address failures identified by various indicators (e.g., poor optimization). We evaluate across various AutoAttack variants by (a) modifying the loss function: cross entropy (CE) and difference of logits ratio (DLR), (b) applying expectation over transformations (EoT), and (c) using random starts. For FMNIST, we use DLR, CE, DLR+EoT, and CE+EoT. For UTKFACE, we use CE and CE+EoT, as DLR is not applicable for binary classification. We report the best attack (least $\phi_{robacc}$).

**Outlier Removal (PoisnRob).** We compute accuracy on $\mathcal{D}_{bd}$ (from adding backdoors to records in $\mathcal{D}_{te}$):

$$\phi_{\text{ASR}}(f_{\text{PoisnRob}}, \mathcal{D}_{\text{bd}}) = \frac{1}{|\mathcal{D}_{\text{bd}}|} \sum_{(x,y) \in \mathcal{D}_{\text{bd}}} \mathbb{I}\left\{\hat{f}_{\text{PoisnRob}}(x) = y\right\}$$

where $y_t$ is the target label chosen by $\mathcal{A}dv$. Ideally, $\phi_{\text{ASR}}$ should be zero. We use BadNets (Gu et al., 2017) to generate poisons by adding a white patch of size 5x5 to the images, applied to 10% of $\mathcal{D}_{tr}$. For POISNROB.IN (Fine-tuning), we fine-tune the last layers of $f$ using random sample of 10% of $\mathcal{D}_{tr}$ without poisons (Sha et al., 2022). For POISNROB.POST (Pruning), we use the implementation from Zheng et al. (2022b). We sweep pruning thresholds from 0.6 to 1.5, in increments of 0.05, to get a model with highest $\phi_{\text{u}}$ and lowest $\phi_{ASR}$.

**Model-Watermarking (MdlWM).** We use the accuracy on $\mathcal{D}_{wmM}$ which is obtained by adding watermarks to data records in $\mathcal{D}_{te}$. We compute this *watermark accuracy* as:

$$\phi_{\text{wmacc}}(f_{\text{MdlWM}}, \mathcal{D}_{\text{wmM}}) = \frac{1}{|\mathcal{D}_{\text{wmM}}|} \sum_{(x,y) \in \mathcal{D}_{\text{wmM}}} \mathbb{I}\left\{\hat{f}_{\text{MdlWM}}(x) = y\right\}$$

where $y_m$ represents the target labels for watermarked records. Ideally, $\phi_{\text{wmacc}}$ should be 100% if the model is successfully watermarked. For MDLWM.PRE (Backdoor), we use BadNets (Gu et al., 2017), similar to Szyller and Asokan (Szyller & Asokan, 2023), by adding a white patch of size 5x5 to 10% of the images in $\mathcal{D}_{tr}$. For MDLWM.IN (Modifying Loss), we use the certified neural network watermarking implementation by Bansal et al. (2022). For MDLWM.POST (API), we use DAWN (Szyller et al., 2021), which flips a fraction of the predictions from target model as watermarks, which is later used to train the surrogate model. Following the original work (Szyller et al., 2021), we apply the watermark to 0.2% of the predictions. Unlike other watermarking schemes, we compute $\phi_{\text{wmacc}}$ on the surrogate model and not the target model.

**Fingerprinting (Fngrprnt.Post).** We use dataset inference (Maini et al., 2021) as our fingerprinting scheme which extracts feature embeddings from $f$, and trains a classifier to distinguish between $\mathcal{D}_{tr}$ and $\mathcal{D}_{te}$. A model is considered stolen if the distance of its embeddings is similar to $f$ with high confidence, and verification is successful if the p-value $< 0.05$. We use $\phi_{\text{pval}}$ as the metric following Szyller and Asokan (Szyller & Asokan, 2023). We use the step size of 1.0 for $L_1$ attack, 0.01 for $L_2$ attack, and 0.001 for $L_{inf}$, and 50 samples for computing p-value from the confidence regressor model.

**Data-Watermarking (DtWM.Pre).** To determine if a dataset was used to train a model, we compare the posterior probability of 100 watermarked testing samples against 100 benign ones using a pairwise t-test (Li et al., 2020). We then calculate the rate of successful detection ($\phi_{\mathrm{rsd}}$), which reflects the percentage of correctly identified watermarked samples from $\mathcal{D}_{\mathrm{wmD}}$ ($\mathcal{D}_{te}$ with watermarks). Watermarks are generated using BadNets (Gu et al., 2017) where 10% of $\mathcal{D}_{tr}$ is watermarked, and we use verification code from Li et al. (2020) to compute $\phi_{\mathrm{rsd}}$. Ideally, $\phi_{\mathrm{rsd}}$ should be 100% for watermarked models.

**Differential Privacy (DiffPriv.In).** We use $\phi_{\mathrm{dp}} = \epsilon_{dp}$, following Szyller and Asokan (Szyller & Asokan, 2023), where ideally, we want a low $\epsilon_{dp}$. We use the implementation from Opacus library (Yousefpour et al., 2021) with a noise multiplier of 1.0 and gradient norm clipping of 1.0 as used in their tutorial for `MNIST`.

**Group Fairness (GpFair.In).** We measure fairness using the *equalized odds gap* on $\mathcal{D}_{te}$ for sensitive attributes $S$ and model predictions $\hat{Y}$, given by: $\phi_{\mathrm{eqodd}}^{\mathrm{GpFair}} = P(\hat{Y} = \hat{y}|S = 0, Y = y) - P(\hat{Y} = \hat{y}|S = 1, Y = y)$ $\forall (\hat{y}, y) \in \{0, 1\}^2$ where an ideal value of zero indicates perfect fairness. For GpFair.In (Regularization), we use code from the fair fairness benchmark that adds a regularization term to penalize equalized odds violations (Han et al., 2023). We set the regularization hyperparameter $\lambda = 1$ which was sufficient to reduce $\phi_{eqodds}$ with $\sim 2\%$ drop in $\phi_{\mathrm{u}}$.

**Explanations (Expl.Post).** We assess the quality fo explanations using *convergence delta*, measures the error between the explanation for a data records and a baseline (Kokhlikyan et al., 2020). We report the average convergence delta across all $\mathcal{D}_{te}$ records as $\phi_{err}$. We use DeepLift (Shrikumar et al., 2017) from Captum library which recommends using a zero vector as a baseline.

# 7 Evaluation

We evaluate individual defenses (§7.1), compare DEF\CON to naïve technique (§7.2 and §7.3), study the impact of hyperparameter tuning (§7.4), and confirm that DEF\CON meets all requirements (§7.5).

## 7.1 Evaluating Individual Defenses

We evaluate the effectiveness of each defense by comparing the metrics $\phi_{(.)}^{\mathrm{D}}$ to a "no defense" baseline. We report the results in Table 3. We also report $\phi_{\mathrm{u}}$ to provide context but do not use it to evaluate accuracy of the technique.

We find that all the defense effectiveness metrics are better than the "no defense" baseline. Once the defenses are applied, we use their respective $\phi_{(.)}^{\mathrm{D}}$ as the "single defense" baseline to compare the effectiveness of the defense combinations later in §7.3. For $\phi_{\mathrm{err}}^{\mathrm{EXPL.POST}}$, we do not have a "no defense" baseline to compare with. Assuming $\phi_{\mathrm{err}}^{\mathrm{EXPL.POST}}$ is effective, we use it as the "single defense" baseline.

## 7.2 Accuracy: using Prior Work

Before empirically evaluating 48 defense combinations, we first identify the combinations which have

Table 3: **Effectiveness of defenses.** For metrics, we use ↑ (resp. ↓) where higher (lower) value is better, and "x" is shorthand for $\phi_{(x)}^{(.)}$. ($\phi_{\mathrm{u}}$ is for context).

| Defense | Metric | FMNIST | UTKFACE |
|---|---|---|---|
| No Defense | **u** (↑) | 90.97 ± 0.18 | 80.28 ± 1.26 |
| | **robacc** (↑) | 7.96 ± 1.24 | 0.00 ± 0.00 |
| | **ASR** (↓) | 99.95 ± 0.04 | 99.98 ± 0.05 |
| | **wmacc.Pre** (↑) | 9.98 ± 0.28 | 0.00 ± 0.00 |
| | **wmacc.In** (↑) | 6.28 ± 1.20 | 62.21 ± 6.03 |
| | **wmacc.Post** (↑) | 0.00 ± 0.00 | 13.33 ± 6.32 |
| | **RSD** (↑) | 0.00 ± 0.00 | 0.00 ± 0.00 |
| | **eqodds** (↓) | | 28.10 ± 6.34 |
| | **dp** (↓) | ∞ | ∞ |
| EvsnRob.In | **u** (↑) | 86.93 ± 0.23 | 73.38 ± 1.15 |
| | **robacc** (↑) | 76.59 ± 0.28 | 37.38 ± 1.30 |
| PoisnRob.In | **u** (↑) | 89.38 ± 0.28 | 79.02 ± 0.30 |
| | **ASR** (↓) | 9.94 ± 0.24 | 56.62 ± 37.83 |
| PoisnRob.Post | **u** (↑) | 86.48 ± 2.35 | 65.42 ± 3.27 |
| | **ASR** (↓) | 66.44 ± 21.30 | 8.59 ± 16.41 |
| MdlWM.Pre | **u** (↑) | 90.15 ± 0.27 | 79.79 ± 0.39 |
| | **wmacc** (↑) | 99.91 ± 0.05 | 100.00 ± 0.00 |
| MdlWM.In | **u** (↑) | 80.87 ± 0.88 | 66.71 ± 10.19 |
| | **wmacc** (↑) | 85.61 ± 2.50 | 93.74 ± 11.00 |
| MdlWM.Post | **u** (↑) | 90.56 ± 0.34 | 80.82 ± 0.45 |
| | **wmacc** (↑) | 100.00 ± 0.00 | 78.10 ± 9.33 |
| DtWM.Pre | **u** (↑) | 90.31 ± 0.27 | 79.93 ± 0.37 |
| | **RSD** (↑) | 100.00 ± 0.00 | 100.00 ± 0.00 |
| Fngrprnt.Post | **u** (↑) | No change | No change |
| | **pval** (↓) | < 0.05 | < 0.05 |
| DiffPriv.In | **u** (↑) | 86.82 ± 0.11 | 74.07 ± 0.28 |
| | **dp** (↓) | $\epsilon_{dp} = 1.36$ | $\epsilon_{dp} = 2.89$ |
| GpFair.In | **u** (↑) | | 76.85 ± 1.99 |
| | **eqodds** (↓) | | 10.89 ± 2.84 |
| Expl.Post | **u** (↑) | No change | No change |
| | **err** (↓) | 0.12 ± 0.03 | 0.59 ± 0.05 |

been empirically evaluated in prior work (§4 and Table 5). We identify eight combinations (**C1-C8**) whose results can be used as ground truth to compare the predictions of DEF\CON and the naïve technique (marked as Ξ or Ξ in Table 5). We use green and red to indicate alignment and conflict among defenses, respectively.

The prediction from a technique is accurate when $\Delta$ (or $\Delta$) for Def\Con, or $\Psi$ (or $\Psi$) for naïve technique, match $\Xi$ (or $\Xi$) taken from prior work (§4) as ground truth. We present additional details to make predictions in **S2**, **S3**, and **S4** using Def\Con in Table 4.

- **C1 (GpFair.Pre + DiffPriv.Pre)** can be combined effectively in the pre-training stage ($\Xi$) (Yaghini et al., 2023). Naïve technique predicts $\Psi$ (same stage) while Def\Con predicts $\Delta$ (defenses make local changes in **S2**).
- **C2 (EvsnRob.In +Fngrprnt.Post)** can be effectively combined ($\Xi$) (Szyller & Asokan, 2023). Naïve technique predicts $\Psi$ (different stages) while Def\Con predicts $\Delta$ (**S3**=no).
- **C3 (DiffPriv.In + Fngrprnt.Post)** can be effectively combined ($\Xi$) (Szyller & Asokan, 2023). Naïve technique predicts $\Psi$ (different stages) while Def\Con predicts $\Delta$ (**S3**=no).
- **C4 (MdlWM.Pre +EvsnRob.In)** are not effectively combined ($\Xi$) (Szyller & Asokan, 2023). Naïve technique predicts $\Psi$ (different stages) while Def\Con predicts $\Delta$ (EvsnRob.In makes poisons ineffective via unintended interaction in **S4**).
- **C5 (DtWM.Pre +EvsnRob.In)** cannot be effectively combined ($\Xi$) (Szyller & Asokan, 2023). Naïve technique predicts $\Psi$ (different stages) while Def\Con predicts $\Delta$ (EvsnRob.In makes poisons ineffective via unintended interaction in **S4**).
- **C6 (MdlWM.Pre + DiffPriv.In)** cannot be effectively combined ($\Xi$) (Szyller & Asokan, 2023). Similar to **C5**, Def\Con predictions this as $\Delta$ while the naïve technique predicts $\Psi$.
- **C7 (DtWM.Pre + DiffPriv.In)** can be effectively combined ($\Xi$) (Szyller & Asokan, 2023). Naïve technique predicts $\Psi$ (different stages) while Def\Con predicts $\Delta$ (DiffPriv.In reduces effectiveness of poisons via unintended interaction in **S4**). Unlike backdoor-based watermarks used in our work, adversarial example-based watermarks used by Szyller and Asokan (Szyller & Asokan, 2023), are inliers which are not suppressed by DiffPriv.In. Hence, Def\Con's prediction differs.
- **C8 (DiffPriv.In + Expl.Post)** can be effectively combined ($\Xi$) (Patel et al., 2022). Naïve technique predicts $\Psi$ (different stages) and Def\Con predicts $\Delta$ (**S3**=no).

*Of the eight combinations,* **Def\Con** *predicts seven correctly, while naïve technique predicts four. This gives a balanced accuracy of 90% (TP=4, TN=3, FP=0, FN=1) for* **Def\Con***, and 40% (TP=4, TN=0, FP=3, FN=1) for the naïve technique.*

Table 4: For each defense, we identify key parameters used in evaluating combination effectiveness (Figure 1): type of change in **S2** (Global, Local, or None); if defense uses a risk in **S3** ("Yes" for backdoors or adversarial examples); and if defense protects against risk in **S4**.

| Defense | S2 | S3 | S4 |
|---|---|---|---|
| EvsnRob.In | Global | No | Yes |
| PoisnRob.In | Global | No | Yes |
| PoisnRob.Post | Global | No | Yes |
| MdlWM.Pre | Local | Yes | No |
| MdlWM.In | Global | Yes | No |
| MdlWM.Post | Local | No | No |
| DtWM.Pre | Local | Yes | No |
| Fngrprnt.Post | None | No | No |
| DiffPriv.In | Global | No | Yes |
| GpFair.In | Global | No | No |
| Expl.Post | None | No | No |

## 7.3 Accuracy: via Empirical Evaluation

We now empirically evaluate the remaining, previously unexplored, combinations to obtain the ground truth and then compute the accuracy of the predictions from both techniques. After removing the eight combinations from prior work, we are left with 40 combinations. We also remove ten combinations where both defenses are applied during in-training. Here, both Def\Con and the naïve technique predict $\Delta$ and $\Psi$ respectively. To apply both defenses in the in-training phase, they can be modified for an effective combination (marked as **T1** in §3). However, this makes it invasive (violates **R3**). Alternatively, defenses can be combined sequentially (e.g., pre-training on the first defense, then fine-tuning on the second), or by alternating the training of both defenses every few epochs. Since, these are non-standard approaches to apply existing defenses, we leave a comprehensive evaluation of ten combinations as future work. Hence, ***we are left with 30 combinations (C9-C38) for empirical evaluation.***

**Predictions from Techniques.** Before evaluating 30 combinations, we denote the defenses as $D_1$ and $D_2$ based on the order in which they are applied. We obtain predictions from Def\Con and the naïve techniques, and indicate them as a tuple: (Naïve prediction, Def\Con prediction). These are indicated in Table 5. We use the information in Table 4 to make predictions in **S2**-**S4** for Def\Con.

Table 5: **Evaluating combinations ($\hat{\mathrm{D}}$):** For brevity, `"x"` in **Metric** column is shorthand for $\phi_{(x)}^{\hat{\mathrm{D}}}$. We use ↑ (resp. ↓) to indicate if a higher (lower) value is better. For defense effectiveness, we use green when $\phi_{(.)}^{\hat{\mathrm{D}}}$ is better or equal to "single defense" baseline; orange for better or equal to "single defense" but worse than "no defense"; red for worse than "no defense". For technique predictions, we use symbol $\Delta$ (resp. $\Psi$) to refer to DEF\CON (naïve technique) and a color code green (resp. red) to indicate alignment (conflict) among defenses. $\mathrm{D}_1$ and $\mathrm{D}_2$ indicate order of applying defenses. ($\phi_\mathrm{u}$ is for context).

| | Combinations | Metric | FMNIST | UTKFACE |
|---|---|---|---|---|
| **C9** | $\mathrm{D}_1$: **EvsnRob.In** $\mathrm{D}_2$: **MdlWM.Post** $(\Psi, \Delta)$ | u (↑) wmacc (↑) robacc (↓) | 90.38 ± 0.22 100.00 ± 0.00 80.43±0.85 | 72.79 ± 0.53 80.95 ± 7.13 42.00 ± 0.49 |
| **C10** | $\mathrm{D}_1$: **PoisnRob.In** $\mathrm{D}_2$: **Fngrprnt.Post** $(\Psi, \Delta)$ | u (↑) ASR (↓) pval (↓) | 89.50 ± 0.21 9.94 ± 0.22 <0.05 | 79.25 ± 1.06 56.09 ± 12.98 <0.05 |
| **C11** | $\mathrm{D}_1$: **PoisnRob.Post** $\mathrm{D}_2$: **Fngrprnt.Post** $(\Psi, \Delta)$ | u (↑) ASR (↓) pval (↓) | 84.73 ± 1.72 61.36 ± 23.96 <0.05 | 63.70 ± 3.87 0.02 ± 0.03 <0.05 |
| **C12** | $\mathrm{D}_1$: **EvsnRob.In** $\mathrm{D}_2$: **Expl.Post** $(\Psi, \Delta)$ | u (↑) err (↓) robacc (↑) | 87.10 ± 0.21 0.22 ± 0.01 79.00 ± 0.21 | 73.65 ± 1.21 0.15 ± 0.04 39.27 ± 0.68 |
| **C13** | $\mathrm{D}_1$: **GpFair.In** $\mathrm{D}_2$: **PoisnRob.Post** $(\Psi, \Delta)$ | u (↑) ASR (↓) eqodds (↓) | | 66.73 ± 3.24 20.21 ± 39.90 2.72 ± 3.20 |
| **C14** | $\mathrm{D}_1$: **MdlWM.Pre** $\mathrm{D}_2$: **GpFair.In** $(\Psi, \Delta)$ | u (↑) wmacc (↑) eqodds (↓) | | 79.02 ± 0.40 98.88 ± 2.13 0.00 ± 0.00 |
| **C15** | $\mathrm{D}_1$: **GpFair.In** $\mathrm{D}_2$: **MdlWM.Post** $(\Psi, \Delta)$ | u (↑) wmacc (↑) eqodds (↓) | | 76.95 ± 1.94 80.95 ± 0.00 7.87 ± 4.72 |
| **C16** | $\mathrm{D}_1$: **DtWM.Pre** $\mathrm{D}_2$: **GpFair.In** $(\Psi, \Delta)$ | u (↑) RSD (↑) eqodds (↓) | | 78.97 ± 1.21 100.00 ± 0.00 0.00 ± 0.00 |
| **C17** | $\mathrm{D}_1$: **GpFair.In** $\mathrm{D}_2$: **Fngrprnt.Post** $(\Psi, \Delta)$ | u (↑) pval (↓) eqodds (↓) | | 78.67 ± 1.46 0.68 ± 0.21 7.46 ± 5.43 |
| **C18** | $\mathrm{D}_1$: **GpFair.In** $\mathrm{D}_2$: **Expl.Post** $(\Psi, \Delta)$ | u (↑) err (↓) eqodds (↓) | | 80.52 ± 0.44 0.16 ± 0.06 12.62 ± 4.20 |
| **C19** | $\mathrm{D}_1$: **PoisnRob.In** $\mathrm{D}_2$: **MdlWM.Post** $(\Psi, \Delta)$ | u (↑) wmacc (↑) ASR (↓) | 89.53 ± 0.36 100.00 ± 0.00 10.48 ± 0.46 | 79.00 ± 0.56 69.52 ± 6.46 38.90 ± 38.73 |
| **C20** | $\mathrm{D}_1$: **MdlWM.Post** $\mathrm{D}_2$: **Expl.Post** $(\Psi, \Delta)$ | u (↑) wmacc (↑) err (↓) | 90.93 ± 0.18 100.00 ± 0.00 0.11 ± 0.02 | 80.53 ± 0.23 72.38 ± 3.56 0.55 ± 0.02 |
| **C21** | $\mathrm{D}_1$: **DtWM.Pre** $\mathrm{D}_2$: POISNROB.IN $(\Psi, \Delta)$ | u (↑) ASR (↓) RSD (↑) | 89.46 ± 0.32 10.18 ± 0.40 0.00 ± 0.00 | 79.00 ± 0.67 77.39 ± 35.23 80.00 ± 40.00 |
| **C22** | $\mathrm{D}_1$: **DtWM.Pre** $\mathrm{D}_2$: **MdlWM.In** $(\Psi, \Delta)$ | u (↑) wmacc (↑) RSD (↑) | 84.45 ± 0.56 89.25 ± 3.48 100.00 ± 0.00 | 79.88 ± 0.27 99.98 ± 0.03 100.00 ± 0.00 |
| **C23** | $\mathrm{D}_1$: **DtWM.Pre** $\mathrm{D}_2$: **PoisnRob.Post** $(\Psi, \Delta)$ | u (↑) ASR (↓) RSD (↑) | 82.90 ± 2.06 64.55 ± 21.23 80.00 ± 40.00 | 69.02 ± 1.96 0.01 ± 0.01 20.00 ± 40.00 |

| | Combinations | Metric | FMNIST | UTKFACE |
|---|---|---|---|---|
| **C24** | $\mathrm{D}_1$: **MdlWM.Pre** $\mathrm{D}_2$: **Expl.Post** $(\Psi, \Delta)$ | u (↑) err (↓) wmacc (↑) | 90.18 ± 0.21 0.14 ± 0.04 99.93 ± 0.06 | 79.76 ± 0.63 0.02 ± 0.03 99.96 ± 0.08 |
| **C25** | $\mathrm{D}_1$: **MdlWM.In** $\mathrm{D}_2$: **Expl.Post** $(\Psi, \Delta)$ | u (↑) err (↓) wmacc (↑) | 86.94 ± 0.50 0.19 ± 0.07 98.24 ± 0.66 | 72.16 ± 5.13 0.37 ± 0.18 97.60 ± 3.54 |
| **C26** | $\mathrm{D}_1$: **DtWM.Pre** $\mathrm{D}_2$: **Expl.Post** $(\Psi, \Delta)$ | u (↑) err (↓) RSD (↑) | 90.04 ± 0.60 0.10 ± 0.04 100.00 ± 0.00 | 79.03 ± 1.10 0.54 ± 0.01 100.00 ± 0.00 |
| **C27** | $\mathrm{D}_1$: **PoisnRob.In** $\mathrm{D}_2$: **Expl.Post** $(\Psi, \Delta)$ | u (↑) ASR (↓) err (↓) | 89.39 ± 0.24 9.79 ± 0.15 0.06 ± 0.02 | 78.71 ± 0.20 44.35 ± 30.07 0.47 ± 0.02 |
| **C28** | $\mathrm{D}_1$: **PoisnRob.Post** $\mathrm{D}_2$: **Expl.Post** $(\Psi, \Delta)$ | u (↑) ASR (↓) err (↓) | 84.62 ± 3.56 76.11 ± 15.85 0.08 ± 0.01 | 63.80 ± 3.37 0.00 ± 0.00 0.15 ± 0.06 |
| **C29** | $\mathrm{D}_1$: **Fngrprnt.Post** $\mathrm{D}_2$: **Expl.Post** $(\Psi, \Delta)$ | u (↑) pval (↓) err (↓) | 90.56 ± 0.16 <0.05 0.11 ± 0.02 | 80.42 ± 0.59 <0.05 0.50 ± 0.03 |
| **C30** | $\mathrm{D}_1$: **DtWM.Pre** $\mathrm{D}_2$: **Fngrprnt.Post** $(\Psi, \Delta)$ | u (↑) pval (↓) RSD (↑) | 90.19 ± 0.59 <0.05 100.00 ± 0.00 | 79.80 ± 0.48 <0.05 100.00 ± 0.00 |
| **C31** | $\mathrm{D}_1$: **DiffPriv.In** $\mathrm{D}_2$: **MdlWM.Post** $(\Psi, \Delta)$ | u (↑) wmacc (↑) dp (↓) | 86.83 ± 0.20 100.00 ± 0.00 $\epsilon = 1.36$ | 74.62 ± 0.49 79.05 ± 3.81 $\epsilon = 2.89$ |
| **C32** | $\mathrm{D}_1$: **DtWM.Pre** $\mathrm{D}_2$: **MdlWM.Post** $(\Psi, \Delta)$ | u (↑) RSD (↑) wmacc (↑) | 90.24 ± 0.29 100.00 ± 0.00 100.00 ± 0.00 | 78.94 ± 0.95 100.00 ± 0.00 62.26 ± 3.77 |
| **C33** | $\mathrm{D}_1$: **PoisnRob.Post** $\mathrm{D}_2$: **MdlWM.Post** $(\Psi, \Delta)$ | u (↑) wmacc (↑) ASR (↓) | 85.09 ± 1.94 100.00 ± 0.00 59.48 ± 24.91 | 67.09 ± 2.81 73.33 ± 8.83 40.20 ± 28.82 |
| **C34** | $\mathrm{D}_1$: **DtWM.Pre** $\mathrm{D}_2$: **MdlWM.Pre** $(\Psi, \Delta)$ | u (↑) wmacc (↑) RSD (↑) | 90.31 ± 0.27 99.96 ± 0.0 100.00 ± 0.00 | 78.53 ± 1.75 100.00 ± 0.00 100.00 ± 0.00 |
| **C35** | $\mathrm{D}_1$: **EvsnRob.In** $\mathrm{D}_2$: **PoisnRob.Post** $(\Psi, \Delta)$ | u (↑) robacc (↑) ASR (↓) | 90.39 ± 0.63 46.00 ± 1.02 79.68 ± 10.25 | 80.28 ± 0.39 0.00 ± 0.00 0.00 ± 0.00 |
| **C36** | $\mathrm{D}_1$: **MdlWM.Pre** $\mathrm{D}_2$: **PoisnRob.In** $(\Psi, \Delta)$ | u (↑) ASR (↓) wmacc (↑) | 89.48 ± 0.15 10.18 ± 0.46 10.18 ± 0.46 | 79.20 ± 0.60 46.92 ± 36.92 46.92 ± 36.92 |
| **C37** | $\mathrm{D}_1$: **MdlWM.Pre** $\mathrm{D}_2$: **PoisnRob.Post** $(\Psi, \Delta)$ | u (↑) ASR (↓) wmacc (↑) | 82.86 ± 4.16 71.32 ± 14.11 71.31 ± 14.10 | 64.09 ± 3.09 0.00 ± 0.00 0.00 ± 0.00 |
| **C38** | $\mathrm{D}_1$: **MdlWM.In** $\mathrm{D}_2$: **PoisnRob.Post** $(\Psi, \Delta)$ | u (↑) ASR (↓) wmacc (↑) | 66.68 ± 9.80 58.59 ± 19.22 58.65 ± 19.23 | 73.69 ± 3.01 99.60 ± 0.37 99.73 ± 0.29 |

- For defenses applied in the same stage (**S1**=yes), the naïve technique predicts $\Psi$. We have the following cases to determine the prediction from DEF\CON:
  1. $\mathrm{D}_2$ makes local/no changes (**S2**=no), DEF\CON predicts this as $\Psi$. We mark them as $(\Psi, \Delta)$ which include **C11**, **C20**, **C28**, **C29**, **C33**, and **C34**.
  2. $\mathrm{D}_2$ makes global changes (**S2**=yes), and DEF\CON predicts this as $\Delta$. We mark them as $(\Psi, \Delta)$ but we did not observe any such combinations.

- For defenses applied in different stages (**S1**=no), the naïve technique predicts $\Psi$. We have the following cases to determine the prediction from DEF\CON:
  1. $D_1$ does not use a risk (**S3**=no) and hence, $D_1$ and $D_2$ do not conflict. We mark them as ($\Psi$, $\triangle$): **C9**, **C10**, **C12**, **C13**, **C15**, **C17-C19**, **C27**, **C31**, and **C35**.
  2. $D_1$, such as MDLWM.PRE and DTWM.PRE, uses a risk (**S3**=yes), but $D_2$ does not protect against this risk (**S4**=no). Hence, there is no conflict and we mark such combinations as ($\Psi$, $\triangle$) which include **C14**, **C16**, **C22**, **C24-C26**, **C30**, and **C32**.
  3. $D_1$, such as MDLWM.PRE and DTWM.PRE, uses a risk (**S3**=yes), and $D_2$ mitigates these risks (e.g., POISNROB). There is a conflict and we mark them as ($\Psi$, $\triangle$) which include **C21**, **C23**, **C36-C38**.

We evaluate the 30 combinations on `FMNIST` and `UTKFACE` (Table 5). For each combination, we compare the effectiveness metrics for each defense to "single defense" from Table 3. We use green to indicate that the metrics are better or similar to "single defense"; orange for worse than single defense but better than "no defense"; and red for similar or worse than "no defense". Metrics marked as orange can still be useful since it provides some protection compared to "no defense". We consider the worst case by a treating a combination as a conflict if atleast one dataset has atleast one metric marked as orange or red.

*Of the 30 combinations,* **Def\Con** *predicts 27 correctly, while the naïve method predicts only 18. This gives a balanced accuracy of 81% (TP=22, TN=5, FP=3, and FN=0) for* **Def\Con** *compared to 36% (TP=16, TN=0, FP=8, and FN=6) for the naïve technique.*

> **Takeaway:** By explicitly accounting for reasons underlying conflicts among defenses, DEF\CON achieves higher accuracy than the naïve technique (satisfies **R1**).

### 7.4 Hyperparameter Tuning for Combinations

We check if hyperparameter tuning can resolve conflicts to see if it can turn (i) **false positives** (predicted as aligned, but empirically conflicting) into **true positives**, and (ii) **true negatives** (predicted and confirmed as conflict) into **false negatives**. We exclude **false negatives** (not observed in our evaluation), and **true positives**, correctly predicted and confirmed as aligned (cannot be improved further with hyperparameter tuning). We use grid search and identify various hyperparameter configurations for defenses in conflicting combinations (Table 6).

**Do False Positives turn to True Positives?**
This includes three combinations (**C17**, **C32**, **C35**), and helps investigate DEF\CON errors.

Table 6: Configurations for hyperparameter tuning of defenses in conflicting combinations.

| Defense | Hyperparameter | Values |
|---|---|---|
| **EvsnRob.In** | Regularization | {2, 4, 6 (default), 8} |
| **PoisnRob.Post** | Pruning threshold | 0.6–1.5 (step 0.05) |
| **MdlWM.Pre** | Trigger size | {3×3, 5×5 (default)} |
| | Watermark fraction | {0.1 (default), 0.2, 0.3} |
| **MdlWM.In** | Watermark fraction | {0.1, 0.2, 0.3} |
| | Training noise | {0.5, 0.75, 1.0 (default), 1.25} |
| | Noise step size | {0.05 (default), 0.10, 0.15} |
| **MdlWM.Post** | Watermark fraction | {0.002 (default), 0.01, 0.02} |
| **DtWM.Pre** | Trigger size | {3×3, 5×5 (default)} |
| | Watermark fraction | {0.1 (default), 0.2, 0.3} |
| **GpFair.In** | Regularization | {0.5, 1 (default), 1.5, 2} |
| **Fngrprnt.Post** | Iterations | {25, 50 (default), 75, 100} |
| | # Fingerprints | {100 (default), 150, 200} |

- **C17 (GpFair.In + Fngrprnt.Post)**. We empirically observe a conflict as FNGRPRNT.POST is ineffective ($\phi_{\text{pval}} > 0.05$), and DEF\CON incorrectly predicts the combination as $\triangle$ in **S3**. We explore the following hyperparameters: regularization for GPFAIR.IN, iterations, and number of fingerprints for FNGRPRNT.POST. For each dataset, we have 36 experiments ($= 4 \times 4 \times 3$). None of the experiments alleviated the conflict. Following prior work (Szyller & Asokan, 2023), we speculate that FNGRPRNT.POST is ineffective because it relies on the decision boundary, which shifts significantly after applying GPFAIR.IN.
- **C32 (DtWM.Pre + MdlWM.Post)**. The combination is empirically effective for `FMNIST` but not for `UTKFACE` where $\phi_{wmacc}$ is less than the "single defense" baseline. DEF\CON incorrectly predicts this combination as $\triangle$ in **S4**. We explore the following hyperparameters: (i) trigger size and watermark fraction for DTWM.PRE; (ii) watermark fraction for MDLWM.POST. For `UTKFACE`, we evaluate 18 experiments ($= 2 \times 3 \times 3$). One experiment with 3×3 trigger size (DTWM.PRE), 30% watermarks (DTWM.PRE), and 2% watermarks (MDLWM.POST), we get $\phi_u = 75.98 \pm 0.61$, $\phi_{RSD} = 100.00 \pm 0.00$ (green), and $\phi_{wmacc} = 70.19 \pm 4.61$ (green). Hence, we remove the conflict and the false positive.

- **C35 (EvsnRob.In + PoisnRob.Post)**. Empirically, there is a conflict as EVSNROB.IN is ineffective (poor $\phi_{robacc}$), and DEF\CON incorrectly predicts as $\triangle$ in **S4**. We vary the regularization hyperparameter (EVSNROB.IN), and pruning thresholds (POISNROB.POST). None of the experiments removed the conflict. We speculate that the model parameters memorizing poisons and adversarial examples overlap. Thus, pruning a model (to reduce $\phi_{ASR}$) trained with EVSNROB.IN, also reduces $\phi_{robacc}$, resulting in a conflict.

**Do True Negatives turn to False Negatives?** We evaluate hyperparameter tuning for five combinations (**C21**, **C23**, **C36**, **C37**, and **C38**).

- **C21 (DtWM.Pre + PoisnRob.In)**. We consider trigger size, and watermark fraction for DTWM.PRE. For each dataset, we get six experiments (=2×3). None of them removed the conflict since POISNROB.IN mitigates backdoors for DTWM.PRE.
- **C23 (DtWM.Pre + PoisnRob.Post)**. We tune the same hyperparameters for DTWM.PRE as in **C21**. For POISNROB.POST, we sweep across various pruning thresholds. For each dataset, we get six experiments (= 2 × 3). For FMNIST, trigger size of 3×3 and 30% watermarks, gives $\phi_u$ =82.00 ± 5.50; $\phi_{RSD}$= 100.00 ± 0.00; $\phi_{ASR}$= 59.90 ± 12.81. This is marked as no conflict ( green ). However, there is a conflict for UTKFACE, making the overall combination a conflict.
- **C36 (MdlWM.Pre + PoisnRob.In)**. We tune the same hyperparameters for MDLWM.PRE, as DTWM.PRE in **C21**. For each dataset, we have six experiments (=2×3). None of them removed the conflict since POISNROB.IN mitigates backdoors for MDLWM.PRE.
- **C37 (MdlWM.Pre + PoisnRob.Post)**. We tune the same hyperparameters for MDLWM.PRE, as DTWM.PRE in **C21**. For POISNROB.POST, we sweep across various pruning thresholds. For each dataset, we get six experiments (=2×3). None of them removed the conflict since POISNROB.POST mitigates backdoors for MDLWM.PRE.
- **C38 (MdlWM.In + PoisnRob.Post)**. We tune the fraction of watermarks, training noise, and step size for MDLWM.IN. For POISNROB.POST, we sweep across various pruning thresholds. For each dataset, we have 36 experiments (=3×4×3). None of them removed the conflict since POISNROB.POST mitigates the backdoors for MDLWM.IN.

**Summary.** We find that hyperparameter tuning is useful in two combinations (**C23** and **C32**). For **C32**, we removed the false positive, thereby increasing DEF\CON's balanced accuracy to 86% (from 81%). For **C23**, we could remove conflict for one of the two datasets, but the combination was still marked as a conflict (no additional false negatives).

> **Takeaway:** Hyperparameter tuning for *conflicting combinations* is important to check if it turns (a) false positives to true positives, or (b) true negatives to false negatives.

**Factors for Hyperparameter Tuning Effectiveness.** We identify factors affecting the effectiveness of hyperparameter tuning and highlight how these factors apply to some conflicting combinations:

- **Expressiveness of hyperparameters:** Tuning is only effective if there are hyperparameters that directly influence the conflicting interaction. If the interaction is insensitive to changes in a hyperparameter, tuning will not resolve the conflict. This could be the reason for hyperparameter tuning being ineffective for some combinations (**C21**, **C36**, **C37**, and **C38**).
- **Search Space:** If the search space is narrow, tuning may never find the optimal configuration to resolve conflicts. Conversely, a sufficiently broad search space increases the chance of finding a configuration that decouples the objectives. There is a possibility that tuning did not work for some combinations as our search space was not broad enough (e.g., **C17**, **C32**, **C35**).
- **Fundamental Incompatibility:** Some defenses are fundamentally incompatible and cannot be resolved by tuning. We discuss this further below.
- **Optimization Landscape:** If the loss landscape contains many local minima, tuning may not find the optimal configuration to resolve conflicts. A practitioner can try more sophisticated tuning instead of grid search (e.g., randomized search or Bayesian optimization), to see if it resolves conflicts.

**Limitations.** When tuning fails to resolve a conflict, we cannot conclusively mark the combination as a conflict: despite considering a feasible search space in our evaluation, some conflict-resolving hyperparameters may have been missed given the vast number of possibilities. Also, when a combination is marked as conflict

(e.g., "**S4**=Yes" for **C21**, **C36**, **C37**, and **C38**), it suggests an *empirical incompatibility* among defenses but does not imply a conclusive proof or *fundamental incompatibility*. Establishing fundamental incompatibility requires a theoretical analysis which is left as future work.

### 7.5 Other Requirements

Having shown that the naïve technique does not perform as well as DEF\CON, we discuss how DEF\CON meets the remaining requirements: scalability (**R2**), non-invasive (**R3**), and generality (**R4**).

**Scalability (R2).** None of the prior works have considered more than two defenses. Since DEF\CON allows for applying defenses in three stages of the ML pipeline, it should theoretically support at least three defenses. To illustrate this, we follow the instructions in §5.3 to extend DEF\CON beyond two defenses. We begin with pairwise combinations predicted as effective (marked as $\triangle$ in Table 5), which align with empirical evaluation, and then include additional defenses. We consider five combinations with three defenses each, which should be effectively combines (marked as $\triangle$). We report the results in Table 7 and find that it is indeed possible to effectively combine three defenses using DEF\CON. *Overall, DEF\CON scales to more than two defenses (R2).* These are illustrative examples to show that DEF\CON is scalable to more than two defenses. Our goal was not an exhaustive evaluation of the large number of multi-way combinations, but only to show that effective multi-way combinations– previously unexplored–are possible. A comprehensive evaluation to identify false positives and negatives, is left as future work.

Table 7: **Scalability (R2) of Def\Con to $D_1$, $D_2$, and $D_3$ (in order).** Color coding and notations are same as in Table 5.

| | Combinations | Metric | FMNIST | UTKFACE |
|---|---|---|---|---|
| **C39** | $D_1$: EVSNROB.IN $D_2$: EXPL.POST $D_3$: MDLWM.POST | u (↑) robacc (↑) err (↓) wmacc (↑) | 87.38 ± 0.15 79.37 ± 0.29 0.96 ± 0.14 100.00 ± 0.00 | 74.34 ± 0.72 39.21 ± 0.32 0.17 ± 0.05 73.33 ± 8.83 |
| **C40** | $D_1$: POISNROB.IN $D_2$: EXPL.POST $D_3$: MDLWM.POST | u (↑) ASR (↓) err (↓) wmacc (↑) | 89.47 ± 0.24 9.81 ± 0.12 0.06 ± 0.02 100.00 ± 0.00 | 79.42 ± 0.51 66.74 ± 12.11 0.52 ± 0.04 77.14 ± 11.82 |
| **C41** | $D_1$: POISNROB.POST $D_2$: EXPL.POST $D_3$: MDLWM.POST | u (↑) ASR (↓) err (↓) wmacc (↑) | 89.47 ± 0.24 9.81 ± 0.12 0.06 ± 0.02 100.00 ± 0.00 | 67.04 ± 3.35 1.85 ± 3.39 0.17 ± 0.10 81.90 ± 7.00 |
| **C42** | $D_1$: DTWM.PRE $D_2$: GPFAIR.IN $D_3$: EXPL.POST | u (↑) wmacc (↑) eqodds (↓) err (↓) | | 77.53 ± 1.75 100.00 ± 0.00 0.00 ± 0.00 0.01 ± 0.00 |
| **C43** | $D_1$: DTWM.PRE $D_2$: GPFAIR.IN $D_3$: MDLWM.POST | u (↑) RSD (↑) eqodds (↓) wmacc (↑) | | 79.17 ± 0.93 100.00 ± 0.00 0.00 ± 0.00 73.33 ± 7.12 |
| **C44** | $D_1$: GPFAIR.IN $D_2$: POISNROB.POST $D_3$: EXPL.POST | u (↑) eqodds (↓) ASR (↓) err (↓) | | 69.42 ± 2.09 8.12 ± 4.49 0.13 ± 0.25 0.05 ± 0.02 |

**Non-Invasive (R3).** DEF\CON extends **T2** and hence, inherits the non-invasive requirement. We use existing defenses proposed in the literature without modifying them, and only adapting them to our datasets. *In summary, DEF\CON satisfies R3.*

**General (R4).** DEF\CON identifies a conflict based on (a) relative position of the defenses in ML pipeline, and (b) the mechanisms that underlie them (whether one defense uses a risk that is being protected by a later defense). DEF\CON does not rely on specific defenses. Hence, DEF\CON is likely to be applicable beyond the initial set of defenses we used for evaluating our work (commonly available defenses from the literature). For identifying conflicts of a new defense with others, a practitioner can first identify its position in the ML pipeline, and determine whether it uses a risk defended by a later defense. Since these are independent of any specific defense, DEF\CON is applicable for any defense that we can map to DEF\CON's flowchart (Figure 1). Furthermore, we select specific defense implementations based on their availability (see §6.4). However, other implementations can be used and should not effect our conclusions.

> **Takeaway:** DEF\CON scales beyond two defenses (**R2**), is non-invasive (**R3**), and general (**R4**).

## 8 Discussion, Conclusions, and Future Work

**Note on Model Utility.** So far, we have focused only on *effectiveness*, examining how combining defenses impacts the effectiveness of each individual defense. An additional pre-requisite for deploying a defense combination is whether it negatively impacts *model utility*. We can define a defense combination to be *viable* if it is (a) effective and (b) incurs only a minimal utility drop compared to lowest of the "single defense"

baseline. In Table 5, we observe that all the combinations which DEF\CON predicted as effective are also viable. For **C15**, **C27**, and **C30**, the utility is worse than the "single defense" baseline. These were already flagged as ineffective. We did not observe any combinations which are effective but not viable.

Extending DEF\CON to predict the viability of the combinations is challenging, since it is unclear how a defense impacts model utility. For instance, there can be the following two cases:

- For both defenses, if utility is either better or similar to the "single defense" baseline, it is likely that the combination will have acceptable utility.
- If the utility degrades for both defenses, the combination is likely to have poor utility and hence non-viable. However, it is also possible (as seen in Table 5) that the utility of the combination does not fall below the "minimum utility for single defenses" baseline, even if some, but not all, constituent defenses fall below their respective "no-defense" baseline. It is unclear what mechanisms account for this phenomenon.

Hence, quantifying the impact of individual defenses on utility is an open problem for several defenses (e.g., adversarial training (Zhang et al., 2019; Tsipras et al., 2019; Yang et al., 2020; Pang et al., 2022; Raghunathan et al., 2020) and differential privacy (Jayaraman & Evans, 2019; Ye et al., 2023; Papernot et al., 2021; Tramèr & Boneh, 2020)), and an area of active research. *Viability* can be included as a requirement in §5.2, and extending DEF\CON for viable combinations is left as future work.

**Practical Considerations.** We discuss the impact on other considerations such as computational cost and latency. The computational cost (for training) and the latency (for inference) are the sum of the costs and latencies, incurred by the constituent defenses when applied individually.

- Pre-training, in-training, and some post-training (e.g., pruning a model) defenses incur a reasonable one-time cost, assuming the practitioner has basic resources to train ML models (e.g., GPUs). These do not have any impact on the inference-time latency.
- For some post-training defenses, there is no training cost but incur a per-inference latency for transforming the inputs or outputs.

**Real-world Impact.** Following the experimental setup in prior work, we evaluated DEF\CON in a lab setting and not on real-world models, which was not realistic for our experiments. The real-world impact of successful combinations depends on the individual application. Practitioners have to decide which combination should be applied for a given application: for instance, in credit card approval, robustness, fairness, and explainability are important properties, while in medical diagnosis, privacy may also be essential.

**Learning-based Component for Def\Con.** We can train a model (e.g., decision tree) to predict the type of interaction for a combination. But this requires a lot more data for training (e.g., features covering various defenses and their combinations), than what we can obtain from the eight combinations in prior work. For the current defenses, our heuristic was sufficient to get a reasonable accuracy. As future work, adding a learning-based component is an interesting direction.

**Other Causes Underlying Conflicts.** While we identify two possible reasons underlying conflicts among defenses (§3.3), we do not claim this to be complete. There could be other underlying reasons which can be included in DEF\CON to make it more accurate. One possible reasons could be the a choice of different $l_p$-norm distances for some defenses (e.g., EVSNROB.IN, and adversarial example-based MDLWM.PRE and DTWM.PRE). Prior works have shown that the objectives of obtaining robustness to different $l_p$-norm bounds are conflicting (Tramèr & Boneh, 2019). In case of combinations, Thakkar et al. (2023) show that choosing different amount of noise for watermarking and adversarial training can remove a conflict. We leave the exploration of additional reasons underlying conflicts as future work.

**Other Combination Techniques.** Duddu et al. (2024a) (Table 3) systematize unintended interactions among defenses and risks, categorizing them as increasing, decreasing, or unexplored. An alternative naïve technique could reject combinations where one defense increases the risks mitigated by another. However, this is restrictive and discards several non-conflicting combinations (e.g., EXPL.POST and MDLWM, EVSNROB.IN and FNGRPRNT.POST). Since there are several unexplored interactions in their systematization, it is challenging to applying this naïve technique in our context. Hence, this technique is limited to some combinations, and not applicable to all combinations in the current state.

**Other Dataset Modalities and Models.** We rely on two image datasets previously used by prior work to evaluate most of our defenses, making them a natural starting point. Since DEF\CON 's steps are modality-independent, we conjecture that it can be applied similarly to other data types. Evaluating DEF\CON on new modalities requires implementing corresponding defenses and models (e.g., language models for text). Then, our methodology can be followed: (a) identifying various risks and corresponding defenses in different stages of the ML pipeline, and (b) use the empirical evaluation of combinations as ground truth to verify the accuracy of DEF\CON. This is a substantial undertaking, and hence, left as future work since it may bring out new insights. We tried to extend our existing defense implementations to tabular dataset, and indeed, not all defenses transfer to other data modalities. For example, when adapting our defenses to the CENSUS dataset, only 4 of the 11 implemented defenses were applicable—image-specific techniques like poisoning and watermarking do not apply to tabular data. This gave two valid combinations: (i) evasion robustness + explanations and (ii) group fairness + explanations (others were excluded due to incompatibility). Both combinations aligned with DEF\CON 's predictions.

**Speculating Combinations with Omitted Defenses.** We *speculate* on the omitted defense combinations from §6: EVSNROB.PRE, DIFFPRIV.PRE, GPFAIR.PRE, and GPFAIR.POST. Since EVSNROB.PRE targets adversarial examples and makes local changes to $\mathcal{D}_{tr}$, we expect its combination with other defenses to behave similar to MDLWM.PRE and DTWM.PRE. DIFFPRIV.PRE and GPFAIR.PRE make global changes by transforming all data records in $\mathcal{D}_{tr}$ and should be applied before other defenses, as we expect them to avoid conflicts. GPFAIR.POST makes global changes in post-training stage to all predictions, and the behavior is likely to be similar to POISNROB.POST which also makes global changes to $f$ in post-training stage. Validating these interactions is left for future work. We indicate some steps for future work to validate our speculation. For defenses omitted due to (i) poor effectiveness: further work is required to design more effective defense variants (e.g., pre-training or post-training evasion robustness). (ii) incompatible datasets: set up experiments for the specific dataset where the defenses work well (e.g., tabular datasets instead of image datasets) and evaluate their combinations. We can then combine with other defenses, and evaluate the combinations. We will release our code to combine new defenses with our existing ones.

**Trade-offs among Defenses.** The current version of DEF\CON only outputs alignment and conflict and does not capture trade-offs. It is not clear how to compare the extent of gains/losses for different defenses, and there is no uniform metric that works across all defenses. For instance, 5% loss in one defense may be much worse than 10% loss in another, but there is no clear consensus on which is better. This may be application dependent and has to be determined a practitioner. Hence, we chose to leave this as future work.

**Summary.** ML models must be protected against multiple risks simultaneously, requiring effective combination of defenses. We systematize prior work, identify unexplored combinations, and evaluate limitations of prior techniques. Using insights from our systematization, we present a technique, DEF\CON, which is more accurate than prior work, does not require modifying defenses, scales to more than two defenses, and applies to various defenses.

## Ethical Considerations

We use public datasets and implementations and none of our experiments require IRB approval.

## Broader Impact

Protecting ML models from various risks simultaneously is an important problem, especially in high-stakes domains where failures can have serious societal consequences. Our work advances the field of designing trustworthy ML systems by moving beyond individual risk mitigation—common in current research—to developing techniques to protect against multiple risks. We do not directly address "accountability" in this work, other than providing a way for model owners to assess effectiveness of defense combinations before deploying them. But it can be combined with additional mechanisms designed to ensure accountability in ML pipelines (such as Duddu et al. (2024b)) for responsible deployment of defenses.

## Acknowledgments

This work is supported in part by Intel (in the context of Private AI consortium), and the Government of Ontario (RE011-038). Vasisht is supported by IBM PhD fellowship, David R. Cheriton Scholarship, and Mastercard Cybersecurity and Privacy Excellence Graduate Scholarship. Views expressed in the paper are those of the authors and do not necessarily reflect the position of the funding agencies. We thank Jian Liu (Zheijang University), Cong Wang (City University of Hong Kong), and Sebastian Szyller (Intel Labs) for fruitful discussions on this topic.

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

## A  Summary of Defenses and Notations

We summarize the different defenses and their impact on $\phi_u$ from §3.1 in Table 8.

## B  Formal Analysis of Def\Con

A defense `D` is defined as a tuple $D := (S, C, R, P)$, where:

- $S$ is the stage where $D$ is applied
- $C$ is the type of changes
- $R$ is the risk it uses
- $P$ is the protection scope of D, and $\emptyset \notin P$.

Let $\mathcal{D}$ be the set of all such defense. The input set for DEF\CON is defined as: $X := \{(D_1, D_2)|D_1 \neq D_2, \text{ and } D_1, D_2 \in \mathcal{D}\}$. DEF\CON can be defined as a function: $f : X \to \{0, 1\}$ where $f(D_1, D_2) = 0$ indicates conflict and $f(D_1, D_2) = 1$ indicates alignment. Given the above formal model, we will discuss consistency, soundness, and completeness.

**Consistency** means that DEF\CON never produces contradictory outputs for the same input pair.

**Claim 1.** *There is no pair $(D_1, D_2)$ for which DEF\CON simultaneously classifies both conflict and alignment, i.e. DEF\CON is consistent.*

*Proof.* According to the definition of $f$, There are two rules or conflict cases:

Table 8: **Summary of defenses.** (Column $\phi_u$ indicates impact on utility: "$\vee$" (decrease), "$\sim$" (no effect), "$\wedge$" $\rightarrow$ (increase).)

| Defense | $\phi_u$ | References |
|---|---|---|
| **EvsnRob (Evasion Robustness)** | | |
| • EVSNROB.PRE (Data Augmentation) | $\wedge$ | Yun et al. (2019); Zhang et al. (2018b); DeVries & Taylor (2017); Madry et al. (2018); Rebuffi et al. (2021) |
| • EVSNROB.IN (Adversarial Training) | $\vee$ | Zhang et al. (2019); Cohen et al. (2019); Lecuyer et al. (2019) |
| • EVSNROB.POST (Input Processing) | $\vee$ | Nie et al. (2022); Song et al. (2018); Guo et al. (2018); Das et al. (2017) |
| | $\sim$ | Grosse et al. (2017); Buckman et al. (2018) |
| **PoisnRob (Outlier Robustness)** | | |
| • POISNROB.PRE (Data Augmentation) | $\vee$ | Borgnia et al. (2021); Qiu et al. (2021); Cretu et al. (2008); Paudice et al. (2018; 2019); Jia et al. (2021b; 2019) |
| • POISNROB.IN (Fine-tuning) | $\sim$ | Diakonikolas et al. (2019); Xu et al. (2019b); Liu & Guo (2020); Patrini et al. (2017); Zhu et al. (2023); Liu et al. (2018); Wu & Wang (2021); Li et al. (2017) |
| • POISNROB.POST (Pruning) | $\vee$ | Zheng et al. (2022b;a); Li et al. (2023c) |
| **MdlWM (Watermarking-M)** | | |
| • MDLWM.PRE (Backdoors) | $\sim$ | Adi et al. (2018); Zhang et al. (2018c); Jia et al. (2021a); Uchida et al. (2017) |
| • MDLWM.IN (Optimization) | $\vee$ | Bansal et al. (2022) |
| • MDLWM.POST (API-based) | $\sim$ | Szyller et al. (2021) |
| **Fngrprnt (Fingerprinting)** | | |
| • FNGRPRNT.POST (Fingerprints) | $\sim$ | Cao et al. (2021); Peng et al. (2022); Lukas et al. (2021); Zheng et al. (2022c); Maini et al. (2021) |
| **DtWM (Watermarking-D)** | | |
| • DTWM.PRE (Backdoors) | $\sim$ | Tekgul & Asokan (2022); Sablayrolles et al. (2020); Liu et al. (2022a) |
| **DiffPriv (Differential Privacy)** | | |
| • DIFFPRIV.PRE (Private Data) | $\vee$ | Xie et al. (2018); Torkzadehmahani et al. (2019) |
| • DIFFPRIV.IN (DPSGD) | $\vee$ | Abadi et al. (2016); Papernot et al. (2017) |
| **GpFair (Group Fairness)** | | |
| • GPFAIR.PRE (Fair Data) | $\vee$ | Kamiran & Calders (2011); Calmon et al. (2017); Zemel et al. (2013); Feldman et al. (2015) |
| • GPFAIR.IN (Regularization) | $\vee$ | Celis et al. (2019); Kearns et al. (2018; 2019); Agarwal et al. (2019; 2018); Zhang et al. (2018a); Kamishima et al. (2012) |
| • GPFAIR.POST (Calibration) | $\vee$ | Pleiss et al. (2017); Hardt et al. (2016); Kamiran et al. (2012); Geyik et al. (2019) |
| **Expl (Explanations)** | | |
| • EXPL.POST (Attributions) | $\sim$ | Ismail et al. (2021); Smilkov et al. (2017); Sundararajan et al. (2017); Koh & Liang (2017); Wachter et al. (2017); Selvaraju et al. (2017); Kim et al. (2018) |

- (c.1) If $S_1 = S_2$ and $C_1 = Global$ and $C_2 = Global$, then $f(\mathtt{D}_1, \mathtt{D}_2) = 0$.
- (c.2) If $S_1 \neq S_2$ and $R_1 \in P_2$, then $f(\mathtt{D}_1, \mathtt{D}_2) = 0$.

For alignment case: (a.1) If none of the conflict conditions hold, then $f(\mathtt{D}_1, \mathtt{D}_2) = 1$.

These conditions partition the input space $X$ into disjoint sets: conflict or alignment, with no overlap. Assume that there exists $X_i$ such that $f(X_i) = 0$ and $f(X_i) = 1$. Since $f(X_i) = 1$, according to (a.1), (c.1) and (c.2) do not hold, leading to a contradiction. Therefore, DEF\CON never produces contradictory classifications for the same input and is consistent. $\qquad\square$

**Soundness** ensures that if DEF\CON predicts a combination as effective (aligned), it should indeed be effective in practice.

This assumption is supported by our empirical evaluation, where DEF\CON achieves:

- 90% accuracy on previously studied defense combinations,
- 81% accuracy on novel, unexplored combinations.

The low rates of false positives and false negatives in these evaluations suggest that DEF\CON can reliably predict combination effectiveness, providing practical evidence for its soundness. A perfectly sound technique would rely on comprehensive empirical evaluation which we want to avoid by proposing an easy-to-use (approximate) technique, DEF\CON. Therefore, the soundness of DEF\CON is conditional on the assumption that these rules perfectly represent all conflict and alignment scenarios: (c.1) (c.2) and (a.1). However, since real-world defense interactions can be complex, involving subtle dependencies and emergent behaviors,

DEF\CON may not fully captured reasons for conflict or alignment. Hence, DEF\CON is not perfectly sound and is likely to incur some errors.

**Completeness** suggests that DEF\CON can classify *every possible defense pair* correctly as either conflict or alignment. Similar to soundness, completeness is conditional on the assumption that the classification rules fully capture all conflict and alignment cases. As a partial theoretical guarantee, we can prove that DEF\CON can classify every defense pair in its input domain, i.e., it produces a classification for every pair without leaving any case unclassified. Assuming that the input set $X$ includes all possible distinct defense pairs, DEF\CON partially satisfies completeness by design. Formally,

**Claim 2.** *For every pair $(D_1, D_2) \in X$, DEF\CON's function $f$ produces a classification $f(D_1, D_2) \in \{0, 1\}$ that identifies whether the defenses conflict or align, i.e., no conflict or alignment case is left unclassified. Therefore, if $X$ includes all distinct combinations of defenses, DEF\CON is partially complete.*

*Proof.* By construction, DEF\CON 's classification rules partition the input space $X$ exhaustively and exclusively by (c.1) (c.2) and (a.1). These rules cover all possible defense pairs in $X$ without overlap or ambiguity, ensuring that every pair is assigned a unique classification, DEF\CON is partially complete on $X$. □

The assumption that $X$ includes all possible defense combinations is supported by the fact that all surveyed defense methods can be represented within the DEF\CON framework.

## C  Formal Analysis for Multi-way Combination Algorithm

The algorithm $F$ is for multi-way combinations and iterates through all permutations of defenses, and uses DEF\CON to check for pairwise conflicts:

(m.1) For defenses $D_i \in \mathcal{D}$, $i = 1, ..., n$, $F(D_1, ..., D_n) = 0$ iff $\exists D_i, D_j \in D$, $f(D_i, D_j) = 0$.

**Consistency.** According to the consistency of $f$, assume that there exists $D_i, i = 1, ..., n$, s.t. $F(D_1, ..., D_n) = 0$ and $F(D_1, ..., D_n) = 1$. Since $F(D_1, ..., D_n) = 1$, $\forall D_i, D_j$, we have $f(D_i, D_j) = 1$, leading to a contradiction. Therefore, $F$ is also consistent.

**Soundness and Completeness.** Under the assumption that $f$ is sound and complete, and considering that we take all possible combinations of defenses (as there is no limit on the input defense set of $F$), the soundness and completeness of $F$ is conditional on the assumption that (m.1) fully captures all conflict and alignment cases.

*Soundness.* Since $f$ has some errors and is not perfectly sound, multi-way combinations can indeed introduce conflicts, which will result in false positives or negatives. Hence, the soundness of of multi-way DEF\CON requires comprehensive evaluation similar to pairwise evaluation. In our evaluation, we only want to show that effective multi-way combinations—previously unexplored—are possible. A comprehensive evaluation to check for soundness is left as future work.

