# OpenReview forum: "Combining Machine Learning Defenses without Conflicts"
_TMLR — Accepted by TMLR_

### Review · Reviewer_6Nbb · 2025-06-12

**Summary Of Contributions:**

This paper introduces DEF CON, a novel technique for combining machine learning defenses without conflicts, inspired by the limitations of prior methods. The key contribution is a systematization of existing work on defense combinations, identifying how defenses are combined and interact, and revealing previously unexplored combinations. ECAC also improves the accuracy of existing combination techniques. The new knowledge presented is that DEF CON accurately identifies whether a defense combination is effective, scalable to combine multiple defenses and is generalizable across different defense types. The authors demonstrate that DEF CON achieves high accuracy (90% on prior combinations and 86% on empirically evaluated unexplored combinations) in determining effective defense combinations, offering practitioners an inexpensive and fast way to assess combinations without extensive empirical evaluation.

**Audience:**

Yes

**Broader Impact Concerns:**

The paper discusses protecting ML models against risks to security, privacy, and fairness. If these combined defenses are deployed in high-stakes or safety-critical applications (e.g., autonomous systems, medical diagnostics, financial decision-making), a Broader Impact Statement should consider - Trust and Accountability.

**Claims And Evidence:**

No

**Requested Changes:**

Would Strengthen the Work:

(1) Clarification on Framework Completeness and Broader Applicability: While the authors acknowledge their initial defense list isn't exhaustive, a more explicit discussion of how this impacts the general applicability of DEF CON beyond the evaluated combinations would be valuable. This would strengthen the claim of "generality (R4)". (2) Addressing Speculative Combinations with Omitted Defenses: The paper speculates on the interactions of certain omitted defense combinations. To strengthen the work, the authors should:
 - More clearly delineate the boundary between empirically validated findings and speculative insights.
- Outline a more detailed plan for future work to empirically validate these speculative interactions
(2) While hyper parameter tuning was beneficial in some cases (C23 and C32) and removed a false positive for C32 , many experiments still resulted in conflicts. A more in-depth analysis would be beneficial - Mainly focus on:
 - Explore why tuning successfully resolved conflicts in some cases but not others. What were the distinguishing factors?
- Provide insights into the inherent limitations of hyper parameter tuning for resolving conflicts, even with DEF CON's guidance.
 - Elaborate on whether the inability to resolve conflicts through tuning for certain combinations indicates a fundamental incompatibility between those defenses that ECAC correctly identifies.

**Strengths And Weaknesses:**

Strengths:
(1) Systematic Approach to Defense Combination: The paper offers a systematic survey of existing work on combining ML defenses, classifying them by combination technique and interaction type (conflict or alignment). This systematization is a significant contribution to our field. (2) Identification of Unexplored Combinations: The research identifies previously unexplored combinations of defenses, revealing research gaps and opportunities for new technique design. (3) Introduction of DEF CON: The paper proposes DEF CON, a novel combination technique designed to be accurate, scalable and generalizable. (4) DEF CON provides an inexpensive and fast method for practitioners to determine if existing defenses can be effectively combined without expensive empirical evaluation. This addresses a significant practical need.

Weaknesses:
(1) While the authors state that their framework is extensible and covers all techniques seen in their systematization , they also acknowledge that the initial list of defenses is not exhaustive. A more explicit discussion of how incompleteness in the initial defense list might impact the broader applicability of DEF CON, beyond the evaluated combinations, would be beneficial. (2)  The paper speculates on the interactions of omitted defense combinations (e.g., EVSNROB.PRE, DIFFPRIV.PRE, GPFAIR.PRE, and GPFAIR.POST). While a full evaluation might be outside the scope, acknowledging the speculative nature and perhaps outlining future work to empirically validate these would be beneficial. (3) A deeper analysis of why tuning succeeded in some cases and failed in others would provide more insight into the limitations of tuning for resolving conflicts.

---

### Review · Reviewer_HMru · 2025-06-22

**Summary Of Contributions:**

This paper addresses the challenge of combining multiple ML defenses (for security, privacy, and fairness) without causing negative interactions. The authors survey existing work, identify gaps, and propose Def\Con, a rule-based technique to predict whether defense combinations are effective. Def\Con is accurate, scalable, non-invasive, and general. It outperforms the naïve baseline in predicting effectiveness across 8 known and 30 new defense combinations.

**Audience:**

Yes

**Broader Impact Concerns:**

N/A.

**Claims And Evidence:**

Yes

**Requested Changes:**

Please see the **Weaknesses** section for a detailed discussion of the issues identified in the paper.

To strengthen the submission and increase its impact, the following key improvements are strongly recommended:

- Provide theoretical justification or formal analysis of Def\Con’s rules to ensure robustness and generalizability.
- Extend the method beyond pairwise defense interactions to capture higher-order effects in defense combinations.
- Broaden empirical validation to include diverse datasets and modalities beyond image classification.
- Address practical considerations such as accuracy, computational cost, latency, and interpretability alongside defense effectiveness.
- Consider incorporating adaptive or learning-based components to improve the flexibility and robustness of Def\Con’s heuristics.
- Include evaluation or discussion on the real-world security and performance impact of the proposed defense combinations.

Additionally, minor issues such as inconsistent figure/table labeling and excessive abbreviations should be addressed to improve readability.

**Strengths And Weaknesses:**

### Strength：
1. The paper addresses an important and underexplored problem: how to combine multiple ML defenses without conflicts. The motivation is clear and valuable.
2. The work is comprehensive and covers a wide range of defenses and interaction types, demonstrating significant effort.
3. The experiments are thorough, with extensive evaluations on both known and unexplored defense combinations.
### Weakness:

**Major:**

1. While Def\Con is rule-based and intuitive, it lacks formal theoretical justification or guarantees (e.g., consistency, soundness, or completeness). Without a formal analysis, it remains unclear whether Def\Con’s decision rules can reliably generalize beyond the evaluated combinations or under adversarial conditions.
2. The method evaluates defenses in a pairwise fashion, assuming that higher-order interactions (e.g., between three or more defenses) can be decomposed into pairwise relationships. This assumption is questionable, as some conflicts may emerge only in multi-way combinations due to cumulative effects, shared resources, or emergent behavior. A principled extension beyond pairwise interactions is currently missing.
3. The empirical validation is limited to two image classification datasets (FMNIST and UTKFACE). This restricts the generalizability of conclusions to other ML settings, such as NLP, generative models, multi-modal models, or structured tabular data, where defense interactions may differ significantly.
4. The current evaluation focuses only on whether a defense combination maintains its functional effectiveness. However, many real-world applications require careful utility–robustness trade-offs. Def\Con does not account for degradation in model accuracy, computational cost, latency, or interpretability that may occur even in “non-conflicting” combinations.
5. The logic underlying Def\Con is based on fixed hand-crafted heuristics (e.g., stage matching, global/local impact). These rules are manually designed and may not capture subtle statistical or semantic interactions between defenses. The lack of adaptability or learning undermines robustness in dynamic or unforeseen scenarios.
6. The paper defines a defense as “effective” if it meets certain metric thresholds, but does not validate whether these combinations lead to improved overall model performance or security in a deployed system. This metric-level evaluation might overestimate the practical value of successful combinations.

**Minor:**

1. Several data tables are incorrectly referred to as "Figure" in the main text (e.g., Figure 3 is actually a tabular summary). For clarity and consistency, these should be labeled as "Table" following standard academic conventions.
2. The paper uses too many abbreviations, which makes it difficult to read and follow. Reducing their use and providing a clear notation summary would improve clarity.

---

### Review · Reviewer_yMcG · 2025-07-02

**Summary Of Contributions:**

The manuscript focuses on the protection of the ML models. The authors first survey the existing work on the possible risks concerning the ML models and the defense techniques used to protect the models. Then, they move onto the combination and interaction of the used defenses. Finally, the authors present the novel defense combination technique called DEF\CON.

**Audience:**

Yes

**Claims And Evidence:**

Yes

**Requested Changes:**

All the concerns mentioned above should be addressed to improve the paper.

**Strengths And Weaknesses:**

**Strengths**

* The authors provide an extensive survey on the used defenses for the ML models, their possible combinations & interactions, and the evaluation of these defenses. Additionally, unexplored combinations are also evaluated.

* The manuscript presents DEF\CON technique which provides proper principles on how to combine ML defenses.

* The manuscript evaluates all possible combinations of defenses constructed with respect to DEF\CON.

* The effect of hyperparameter tuning on ML defenses is investigated.

**Weaknesses**

* Experiments are limited to image datasets. Since these defenses are not restricted to the domains, what would be their performance for different domain datasets such as text or tabular data?

* Conflicts are treated as binary classifications, conflict (not effective) and alignment (effective). However, what would be the case for tradeoffs? For the proposed technique, combining two defenses with a little tradeoff on one defense to gain high benefit on another will not be applied since they conflict. Yet, such a combination may be more effective than the other combinations.

* How would DEF\CON be applied automatically? The impact and the stages of the defenses require manual labelling.

* DEF\CON is only compared against the mutually exclusive stage rule. The performance comparison seems to be insufficient.

---

> ### Author Response · Authors · 2025-07-07
> **Clarifications and Proposed Changes**
>
> Thank you for your feedback. We clarify the points you raised (in **bold**) and indicate our proposed changes (in *italics* marked with →). *Please let us know if these responses address your concerns sufficiently.*
>
> **DEF\CON’s accuracy for different domain datasets such as text or tabular data**
>
> We rely on two image datasets previously used by prior work to evaluate most of our defenses, making them a natural starting point. Since DEF\CON’s steps are modality-independent, it can be applied similarly to other data types.
>
> Not all defenses transfer to other data modalities. For example, when adapting our defenses to the CENSUS dataset, only 4 of the 11 implemented defenses were applicable—image-specific techniques like poisoning and watermarking do not apply to tabular data. This yielded two valid combinations: (i) evasion robustness + explanations and (ii) group fairness + explanations.
> - (In-training combinations and differential privacy + explanations were excluded due to incompatibility.)
>
> Both evaluated combinations aligned with DEF\CON’s predictions.
>
> Evaluating DEF\CON on new modalities requires implementing corresponding defenses and models (e.g., language models for text), which is a substantial effort and left for future work, since it may bring out new insights.
>
> → *We will include this discussion in §8*
>
> **Treatment of Trade-offs**
>
> Thank you for raising this point. Indeed, the current version of DEF\CON only outputs alignment and conflict and does not capture trade-offs. It is not clear how to compare the extent of gains/losses for different defenses, and there is no uniform metric that works across all defenses. For instance, 5% loss in one defense may be much worse than 10% loss in another, but there is no clear consensus on which is better. This may be application dependent and has to be determined a practitioner. Hence, we chose to leave this as future work.
>
> → *We will include this discussion in §8*
>
> **How would DEF\CON be applied automatically?**
>
> Automation is not an important criterion for DEF\CON. For each combination the practitioner needs to run Def\Con once (i.e., the frequency of applying DEF\CON is comparable to the frequency of training a model). Additionally, the features required for DEF\CON are easy to identify and well-documented:
> - *Stage of a defense* would be known to the practitioner who decides to use a defense (information available as part of its description)
> - *How a defense interacts with various risks* has been documented extensively. For each defense, we know the risk it protects against (summarized in §3.1), while for interactions with unrelated risks have been systematized in Table 3 of Duddu et al., 2024 (SoK: Unintended Interactions among ML Defenses and Risks).
>
>
> **Only comparison with Mutually Exclusive Placement**
>
> We systematically surveyed prior works to identify two techniques: optimization-based and Mutually Exclusive Placement aka naive technique. Our focus is on combining unmodified defenses since it lowers the bar for practitioners to combine defenses and does not require expert knowledge. Hence, we discarded optimization-based techniques since it does not meet the requirements (§5.2), and it is ad-hoc: optimizations for combining a set of defenses are not applicable to others.
>
> Hence, comparing naive techniques with DEF\CON made sense given the requirements. For accuracy (R1), we comprehensively showed that DEF\CON significantly outperforms the naive technique. Given this, we ruled out the naive technique as a good choice and did not evaluate against the remaining requirements, focusing on DEF\CON.
>
>  *Please let us know if we misinterpreted your question.*

---

### Author Response · Authors · 2025-07-11
**Summary of Changes for the Revised Version**

Thank you for your constructive feedback. We have uploaded the revised version of the paper based on the discussion so far (changes indicated in red). Below, we summarize the proposed changes (in **bold**), and indicate the changes made. *Please let us know if these changes address your concerns sufficiently.*

As the first to systematically explore effective defense combinations, our work offers key insights and evaluation guidelines, along with highlighting several promising directions for future research.

##  Reviewer 6Nbb

**Clarification on Broader Applicability**

We updated  §7.5: General to clarify why DEF\CON is general.

**Future plan to validate speculative interactions**

We updated the text in §8 to clarify the speculative nature, and added next steps to validate them for future work.

**Distinguishing factors for why tuning resolves conflicts in some cases**

We added the relevant text towards the end of §7.4 under a separate heading, speculating the reasons about the factors for combinations where hyperparameter tuning works or not.

**Limitations of hyperparameter tuning + Fundamental incompatibility**

We added the relevant text towards the end of §7.4 under a separate heading.

**Broader Impact Concerns**

We added text on broader impact as a separate heading at the end.



##  Reviewer HMru

**Formal justification (e.g., consistency, soundness, or completeness)**

We have added formal analysis in Appendix B and refer to it in §5.3.

**Multi-way combinations from pairwise combination**

We have added the algorithm for checking conflicts in multi-way combinations from pairwise combinations in §5.3 along with its formal analysis in Appendix C.

**Diverse datasets and modalities**

In §8, we added a heading on extending our evaluation to new datasets and modalities, including new results for applying our defenses to a tabular dataset.

**Practical considerations (accuracy, cost, latency)**/ **Adaptive or learning-based components to improve Def\Con’s heuristics**/ **Real-world security and performance impact of the proposed defense combinations.**

We have added relevant text in §8 under a separate heading.

**Minor Issues**

We have updated the captions which incorrectly indicated tables as figures.


## Reviewer yMcG

**DEF\CON’s accuracy for different domain datasets such as text or tabular data**

In §8, we added a heading on extending our evaluation to new datasets and modalities, including new results for applying our defenses to a tabular dataset.

**Treatment of Trade-offs**

We added text under a separate heading in §8.

---

### Decision · Action_Editor_CraE · 2025-08-12

**Recommendation:** Accept as is

**Audience:**

Yes

**Audience Explanation:**

This paper presents a comprehensive analysis of the combination of multiple defenses in machine learning models, examining their effects and potential conflicts. All reviewers find the results insightful. While some reviewers pointed out that the results are limited to image data, and there were limited evaluations on the utility-robustness trade-offs, the authors did put some discussion of such limitations in the revised version.

**Claims And Evidence:**

Yes

**Claims Explanation:**

This is an empirical study with sufficient experiments for justification.